https://doi.org/10.1038/s41467-021-25430-9　　**OPEN**

# Genetic and epigenetic basis of hepatoblastoma diversity

Genta Nagae [1,12], Shogo Yamamoto [1,12], Masashi Fujita [2,12], Takanori Fujita[1], Aya Nonaka[1], Takayoshi Umeda [1], Shiro Fukuda [1], Kenji Tatsuno [1], Kazuhiro Maejima[2], Akimasa Hayashi [1,3], Sho Kurihara[4], Masato Kojima[4], Tomoro Hishiki[5], Kenichiro Watanabe[6], Kohmei Ida[7], Michihiro Yano[8], Yoko Hiyama[11], Yukichi Tanaka[9], Takeshi Inoue[10], Hiroki Ueda[1], Hidewaki Nakagawa[2,13], Hiroyuki Aburatani [1,13] & Eiso Hiyama [4,11,13 ✉]

Hepatoblastoma (HB) is the most common pediatric liver malignancy; however, hereditary predisposition and acquired molecular aberrations related to HB clinicopathological diversity are not well understood. Here, we perform an integrative genomic profiling of 163 pediatric liver tumors (154 HBs and nine hepatocellular carcinomas) based on the data acquired from a cohort study (JPLT-2). The total number of somatic mutations is precious low (0.52/Mb on exonic regions) but correlated with age at diagnosis. Telomerase reverse transcriptase (*TERT)* promoter mutations are prevalent in the tween HBs, selective in the transitional liver cell tumor (TLCT, > 8 years old). DNA methylation profiling reveals that classical HBs are characterized by the specific hypomethylated enhancers, which are enriched with binding sites for ASCL2, a regulatory transcription factor for definitive endoderm in Wnt-pathway. Prolonged upregulation of ASCL2, as well as fetal-liver-like methylation patterns of *IGF2* promoters, suggests their "cell of origin" derived from the premature hepatoblast, similar to intestinal epithelial cells, which are highly proliferative. Systematic molecular profiling of HB is a promising approach for understanding the epigenetic drivers of hepatoblast carcinogenesis and deriving clues for risk stratification.

[1] Genome Science Laboratory, Research Center for Advanced Science and Technology (RCAST), the University of Tokyo, Tokyo, Japan. [2] Laboratory for Cancer Genomics, RIKEN Center for Integrative Medical Sciences, Yokohama, Japan. [3] Department of Pathology, Kyorin University Faculty of Medicine, Tokyo, Japan. [4] Department of Pediatric Surgery, Hiroshima University Hospital, Hiroshima, Japan. [5] Chiba University Graduate School of Medicine, Chiba, Japan. [6] Shizuoka Children's Hospital, Shizuoka, Japan. [7] Department of Pediatrics, Teikyo University Mizonokuchi Hospital, Kawasaki, Japan. [8] Department of Pediatrics, Akita University Hospital, Akita, Japan. [9] Department of Pathology, Kanagawa Children's Medical Center, Yokohama, Japan. [10] Department of Pathology, Osaka City General Hospital, Osaka, Japan. [11] Department of Biomedical Science, Natural Science Center for Basic Research and Development, Hiroshima University, Hiroshima, Japan 734-8551 1-2-3, Kasumi, Minami-ku, Hiroshima. [12] These authors contributed equally: Genta Nagae, Shogo Yamamoto, Masashi Fujita. [13] These authors jointly supervised this work: Hidewaki Nakagawa, Hiroyuki Aburatani, Eiso Hiyama. ✉email: eiso@hiroshima-u.ac.jp

Hepatoblastoma (HB) is the most common malignant pediatric liver tumor and one of the fastest-rising cancers in children younger than 5 years of age[1]. Hepatocellular carcinoma (HCC) is sometimes seen in older children and adolescents. An interesting hybrid tumor, with HB and HCC, features, variably called transitional liver cell tumor (TLCT) or hepatocellular malignant neoplasm, not-otherwise-specified (HCN-NOS), exists on the age continuum between HB and HCC and is usually treated as HB[2]. HB is thought to be derived from hepatic precursor cells and is morphologically similar to immature hepatocytes. Given that the prognoses of the patients vary widely, tumor distribution, stage of the tumor, and complete tumor resection have been proposed as prognostic indicators in HB[3,4].

Several multi-center trials, such as the Société Internationale d'Oncologie Pédiatrique Epithelial Liver (SIOPEL), Children's Oncology Group (COG), and JPLT (the Japanese study group for Pediatric Liver Tumors) studies have achieved a successful reduction of large HB by preoperative chemotherapy and complete resection[4,5]. In advanced tumors with a low malignant grade, standard chemotherapeutic regimens are effective and result in longer survival, while aggressive chemotherapies, such as molecular targeted therapy, are needed for tumors with a high malignant grade. In Japan, the JPLT-2 study was conducted between 2000–2012, and the 5-year event-free survival (EFS) rate of patients with early stages was more than 80%, but those with advanced stages were under 60%[6,7]. Clinical parameters, such as the age at diagnosis, PRETreatment EXTent (PRETEXT) of disease classification, and serum alpha-fetoprotein (AFP), level have been confirmed as clinical predictors but are imperfect. Therefore, molecular evaluation of the malignant grade of HB is necessary to improve the outcome of patients with highly malignant HB[8].

Several molecular markers have been previously analyzed to identify HB with high malignancy potential, including loss of heterozygosity (LOH) of chromosome *11p15.5*, which is often affected in nephroblastoma and rhabdomyosarcoma (RMS) in children and may contain putative tumor suppressor genes for HB[9] but is unlikely to be a prognostic marker[10,11]. The mutation or deletion of the ß-catenin gene (*CTNNB1*) exon 3 is frequently detected in HB, suggesting activation of the wingless/WNT signal pathway[12]. While this plays an important role in the pathogenesis of HB, it is not considered a predictive molecular marker for distinguishing high-risk tumors from other tumors[11,13]. Very low birth weight is associated with a significantly increased risk of HB, but the underlying mechanism remains unknown[14,15]. Despite these previous findings and recent large-scale genomic analyses[16], HB has been recognized as the tumor with the fewest somatic mutations among all pediatric solid tumors. Therefore, it is necessary to identify the epigenetic drivers and explore the "cell of origin" for this malignancy. Children with Beckwith–Wiedemann syndrome (BWS), especially uniparental disomy (UPD) at 11p15.5, are at increased risk of embryonal tumors including HB. BWS is a congenital overgrowth disorder that is mainly associated with altered genomic imprinting at chromosome 11p15.5[17]. The most common epigenetic defect in BWS is dysregulation at the 11p15.5 imprinting center, which suggests that DNA methylation or imprinting alterations also play important roles in HB development and progression.

In the JPLT-2 study, more than 300 patients were treated following the central pathological diagnosis of tumor samples; the tumor and noncancerous tissues were stored before treatment. In this work, using pre-treatment samples in this clinical trial cases, we elucidate the molecular mechanism during hepatoblast carcinogenesis and identify the useful molecular biomarkers for HB risk stratification by comprehensive genomic, epigenomic, and transcriptomic analyses including whole-genome-sequencing and exome-sequencing, RNA sequencing, SNP array, and DNA methylation array.

## Results

**Samples, clinical data, and analytical approach.** We analyzed 163 histologically confirmed tumors (154 HB and 9 HCC tumors) and matched blood or non-cancerous liver tissue as normal in the JPLT-2 clinical trial cases. The mean age at diagnosis of HB was 25.5 months, and the female/male ratio was 0.74. Primary tumors spanned all PRETEXT (I, 19 cases; II, 47 cases; III, 58 cases; IV, 39 cases), and 63 cases had extrahepatic distension. The median overall survival (OS) of HB cases was 55.0 months, with a 5-year OS of 81.6%. Clinical data, including the age at diagnosis, PRETEXT, tumor histology, and survival, are summarized in Supplementary Table 1, and dataset 1. We generated a comprehensive molecular dataset of the 163 tumors as follows: whole-exome sequencing (WXS, $n = 112$), whole-genome sequencing (WGS, $n = 33$), Affymetrix 6.0 single nucleotide polymorphism (SNP) arrays ($n = 112$), RNA-sequencing ($n = 111$), Illumina Infinium DNA methylation BeadChip ($n = 146$), and whole-genome bisulfite sequencing (WGBS, $n = 20$).

Our comprehensive characterization of molecular aberrations in HB consisted of three main parts. First, we identified somatic single-nucleotide variants, gene fusions, copy-number alterations, and germline variants of the 40 cancer predisposition genes (Supplementary Table 2) to clarify the core genomic events of driver genes as acquired genetic aberrations and hereditary cancer predisposition. Next, to uncover the diverse pathways of hepatocarcinogenesis according to their environmental effects, we performed the methylome profiling and revealed the distinct subtypes, which are related to clinicopathological features, genomic alterations, and gene expression signatures. Lastly, we evaluated the potential of the clinical information and the molecular features for precise stratification of HB patients.

**Landscape of somatic and germline mutation of driver genes in HB.** To identify recurrently mutated genes, we analyzed 112 childhood liver tumors that included HB and HCC using the Karkinos pipeline (https://github.com/genome-rcast/karkinos) as previously reported[18]. Genetic events of somatic mutations in exonic regions were remarkably rare (0.52 per Mb on average), but 26 recurrently mutated genes, including β-catenin (*CTNNB1*), *ARID1A, TERT* (in promoter region), *ITPR2*, and *APC*, were identified in this cohort (Fig. 1a). In addition to the whole-exome analyses, large deletions of *CTNNB1* in exon 3 were detected by long PCR and Sanger sequencing. Consistent with prior reports, *CTNNB1* was the most frequently mutated gene (77.6%). Similar to HCC in adults[18], 35.0% of HB (57 cases) had single base substitution in exon 3 that affects phosphorylation sites for GSK3B. Another 44.8% of HB (73 cases) showed in-frame deletion within or across exons 3 and 4. *TERT* promoter mutations were observed at the same hotspot ($-124$, $-141$ bp from TSS) similar to the other adult cancers that included HCC. The 125 HB cases were divided into three groups according to age at diagnosis: "tween HB (age > 8, 6 cases)", "child HB (age = 2–8, 38 cases)", and "infant HB (age < 2, 81 cases)". Tween HB was characterized by *TERT* promoter-mutations (5 cases, $p = 4.9 \times 10^{-8}$ by Fisher's exact test) and histologic subtype of TLCT or HCN-NOS[19] ($p = 5.3 \times 10^{-7}$). Deletion of *CTNNB1* exon 3 was observed to be significant in child HB ($p < 0.046$). Fetal subtype and BWS were found to be more predominant in infant cases ($p = 0.011$ and $7.6 \times 10^{-7}$, respectively). Although *NFE2L2* mutation was observed in 10% of HBs in a previous

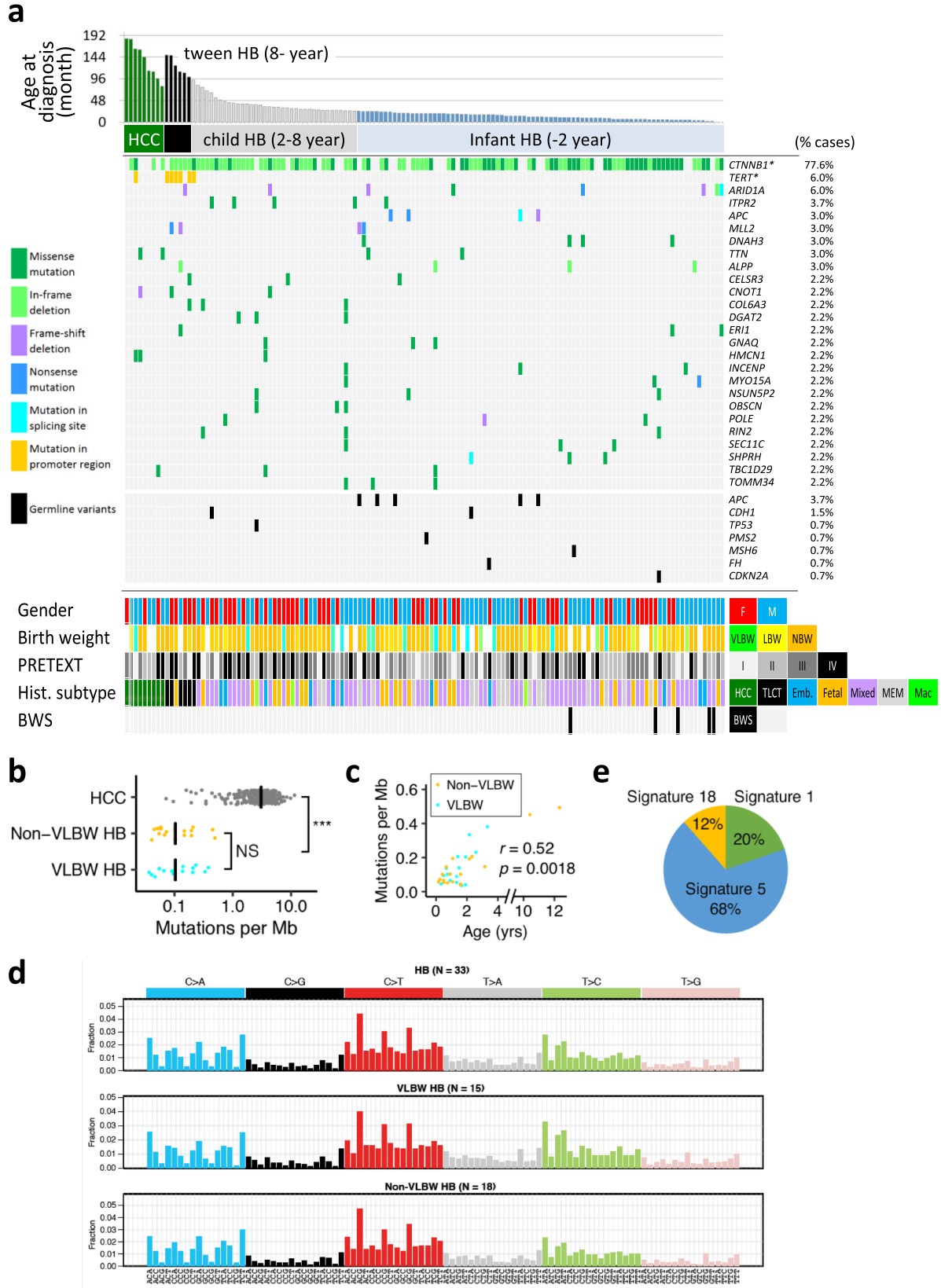

study[20], we did not find cases with *NFE2L2*-mutations in our cohort.

To gain insight into the genetic predisposition to HB, we compiled a list of 40 cancer predisposition genes and examined their germline variants in patients with HB. Truncating or pathogenic germline variants of these genes were found in 9 cases

(6.1%) among 147 WXS/WGS (Supplementary Table 2). Germline mutations of *APC* were most frequent and were found in 5 cases (3 nonsense mutations and 2 frameshift deletions), two of which were already diagnosed as familial adenomatous polyposis (FAP). Germline mutations of *APC* were mutually exclusive with somatic mutations of *CTNNB1* ($p = 0.0003$ by Fisher's exact test).

**Fig. 1 Landscape of driver mutations and mutational spectrum in childhood liver neoplasms. a** A total of 134 cases (WXS: 112, WGS: 33) of childhood hepatoblastoma (HB) and hepatocellular carcinoma (HCC) were sorted by age at diagnosis. The bar graph at the top of the panel shows individual age at diagnosis (childhood HCC, $n = 9$; tween HB (age > 8), $n = 6$; child HB (age 2–8), $n = 38$; infant HB (age < 2), $n = 81$). The middle of the panel shows the driver mutations detected in at least 3 cases in this cohort (missense mutation, dark green; in-frame deletion, light green; frame-shift deletion, purple; nonsense mutation, blue; mutation in splicing site, light blue; mutation in the promoter region, yellow) and the pathogenic germline variants (black). The image at the bottom of the panel shows clinical and pathological information. VLBW, very low birth weight (<1500 g); LBW, low birth weight (1500–2500 g); NBW, normal birth weight (2500–3500 g), HCC hepatocellular carcinoma, TLCT transitional liver cell tumor, Embry. embryonic, Macro macrotrabecular; BWS, Beckwith–Wiedemann Syndrome. **b** The number of somatic mutations per million base pairs of HCC ($n = 269$) and HB with VLBW ($n = 15$) and non-VLBW ($n = 18$) detected by WGS. Statistical significance was analyzed by Wilcoxon rank-sum test. **c** Correlations between the age at diagnosis and the number of somatic mutations of HB with VLBW ($n = 15$) and non-VLBW ($n = 18$). Statistical significance was analyzed by the Spearman correlation test. **d** Signatures of mutational processes extracted from the mutational catalog of 33 HBs with/without VLBW. **e** percentages of the contributing mutational signatures.

Among the five patients with germline *APC* mutations, two patients also exhibited somatic truncating mutations of *APC*. Germline mutations of *TP53* (p.R158H) and *BRCA2* (p.R3128X) were also detected in patients whose tumors had wild-type *CTNNB1*. Analysis of germline variants of uncertain significance revealed germline missense variants in several genes in the Wnt/β-catenin signaling pathway: *APC* (7 cases), *RNF43* (3 cases), *AXIN1* (1 case), *AXIN2* (1 case), and *CTNNB1* (1 case). We also found that four cases had germline mutations in *HNF1A*, which is related to multiple hepatic adenoma development and young-onset diabetes[21,22].

**Mutation burden and signatures according to the age at diagnosis and birth weight.** The WGS that was performed on the 33 HB genomes to confirm the extreme scarcity of somatic mutations revealed that the median number of single nucleotide variants (SNVs) and insertions and deletions (INDELs) at the whole-genome level was 248 and 53, respectively. The median mutation frequency of HB was 0.10 mutations per Mb, which was significantly lower than that of adult HCC (3.06 per Mb, $n = 269$; $p < 3.7 \times 10^{-20}$ by Wilcoxon rank-sum test)[23] (Fig. 1b). Among the 33 patients with HB, 15 cases had very low birth weight (VLBW; birth weight < 1500 g), and 18 cases had non-VLBW. Mutation frequency was comparable between VLBW and non-VLBW HB (both 0.10 per Mb; $p = 0.9$). The mutation frequency of HB was positively correlated with the age at diagnosis (Spearman correlation coefficient 0.52, $p = 0.0018$) (Fig. 1c). Two older cases had the highest mutation frequencies (0.45 and 0.49 per Mb, Supplementary Fig. 2a). However, the positive correlation between the age at diagnosis and mutation frequency was retained even when the tween HB cases were excluded (Spearman correlation coefficient 0.42, $p = 0.018$).

Our analysis of the patterns of somatic base substitutions in the WGS of 33 HB genomes (Fig. 1d) revealed that the most frequent base substitution was the C-to-T transition (32%), followed by the C-to-A transversion (21%), and the T-to-C transition (20%). Trinucleotide substitution patterns were largely similar among 33 HB (Supplementary Fig. 2a), and the patterns in VLBW and non-VLBW resembled each other (cosine similarity = 0.98) (Fig. 1d). The substitution patterns were decomposed into known mutational signatures of COSMIC, revealing a contribution from three signatures: Signature 5 (69%), Signature 1 (20%), and Signature 18 (11%) (Fig. 1e, Supplementary Fig. 2b). Signature 1 and Signature 5 are associated with aging and are found in various adult and childhood cancers[24]. Signature 18 contributes to mutations in several childhood cancers (neuroblastoma, acute myeloid leukemia, and RMS)[16,25]. Recent evidence suggests that Signature 18 is associated with DNA damage by reactive oxygen species[26,27]. However, contributions of signatures 1, 5, 18 were not correlated with the age at diagnosis and the birth weight (Supplementary Fig. 2c, d).

**Copy number alterations, SVs, and genomic stability.** Somatic copy number alterations of 112 HBs were assayed using Affymetrix 6.0 SNP arrays to identify arm-level gains and losses, focal amplifications and deletions, and UPD (Fig. 2a). Arm-level gains were frequent in chromosomes 1q, 2q, 20, 2p, and 6p (40%, 28%, 23%, 21%, and 15%, respectively), whereas arm-level losses were common in chromosomes 1p, 4q, and 11q (16%, 10%, and 10%, respectively). Focal amplifications were found in chromosome 2q24.3 (9%, 10 cases), and focal deletions were observed in chromosomes 4q35.1 (21%, 24 cases) and 5q22.2 (3%, 3 cases). The 4q35.1 deletions contained the *IRF2* gene (Fig. 2b), which is reported as a p53-related tumor suppressor gene in HCC[28]. The 5q22.2 deletions affected *APC*, suggesting an alternative mechanism of Wnt-signaling activation in HB. Among the three patients with focal 5q22.2 deletions, one patient had a germline mutation of *APC*. The deletion affected the wild-type *APC* allele of the patient, thus constituting a somatic second hit to the gene. Allele-specific calling on SNP arrays revealed 39 cases (34.2%) of UPD in chromosome 11p containing the *H19/IGF2* imprinted regions (Fig. 2c, Supplementary Fig. 3). Hierarchical clustering of copy number profiles revealed that 32 cases (29%) had chromosomal instability (CIN) (Fig. 2d). Gain of chromosome 20 was characteristic of CIN tumors and was found in 26 out of 32 cases (81%). Compared to the clinical outcomes, the distant metastasis and postoperative recurrence were significant in the CIN cases ($p = 0.018$, $1.4 \times 10^{-4}$, respectively). As genomically stable (GS) cases appeared to be almost diploid by GISTIC 2.0, the detailed analysis identified recurrent focal amplification of chromosome 2q, focal deletion of chromosome 4, and a high frequency of UPD or LOH of chromosome 11.

WGS analysis identified 152 somatic structural variants (SVs) in 33 HB genomes (Supplementary dataset 2). The median number of somatic SVs per sample was 2.0. The number of somatic SVs was positively correlated with age at diagnosis ($p = 0.0043$ by Jonckheere–Terpstra test). VLBW- and non-VLBW HB had a similar number of SVs ($p = 0.5$ by Wilcoxon rank-sum test). Somatic SVs were enriched in several chromosomal regions (Supplementary Fig. 4b). The most frequently affected regions were 4q34.3–35.2 (9 of 33 cases), 3p22.1 (8 cases), and 2q24.1–3 (6 cases). All of the 3p22.1 SVs were deletions of the exon 3 of the *CTNNB1* gene (Supplementary Fig. 4c). These deletions were not detected by SNP array-based copy number analysis because of their short lengths (36–1,049 bp). Two of the 8 deletions extended further to exon 4 of *CTNNB1*. Enrichment of SVs in 4q34.3–35.2 and 2q24.1–3 is consistent with our observation in copy number analysis. The collation of SV and copy number gave an insight into the loss of *IRF2* in 4q35.1. In case 17, complex rearrangements in 4q led to the homozygous deletion of *IRF2* (Supplementary Fig. 5a). In case 42, SVs between 4p and 4q resulted in LOH of *IRF2* (Supplementary Fig. 5b). Case 98 had tripartite translocations among 1p, 4q, and 11q, which

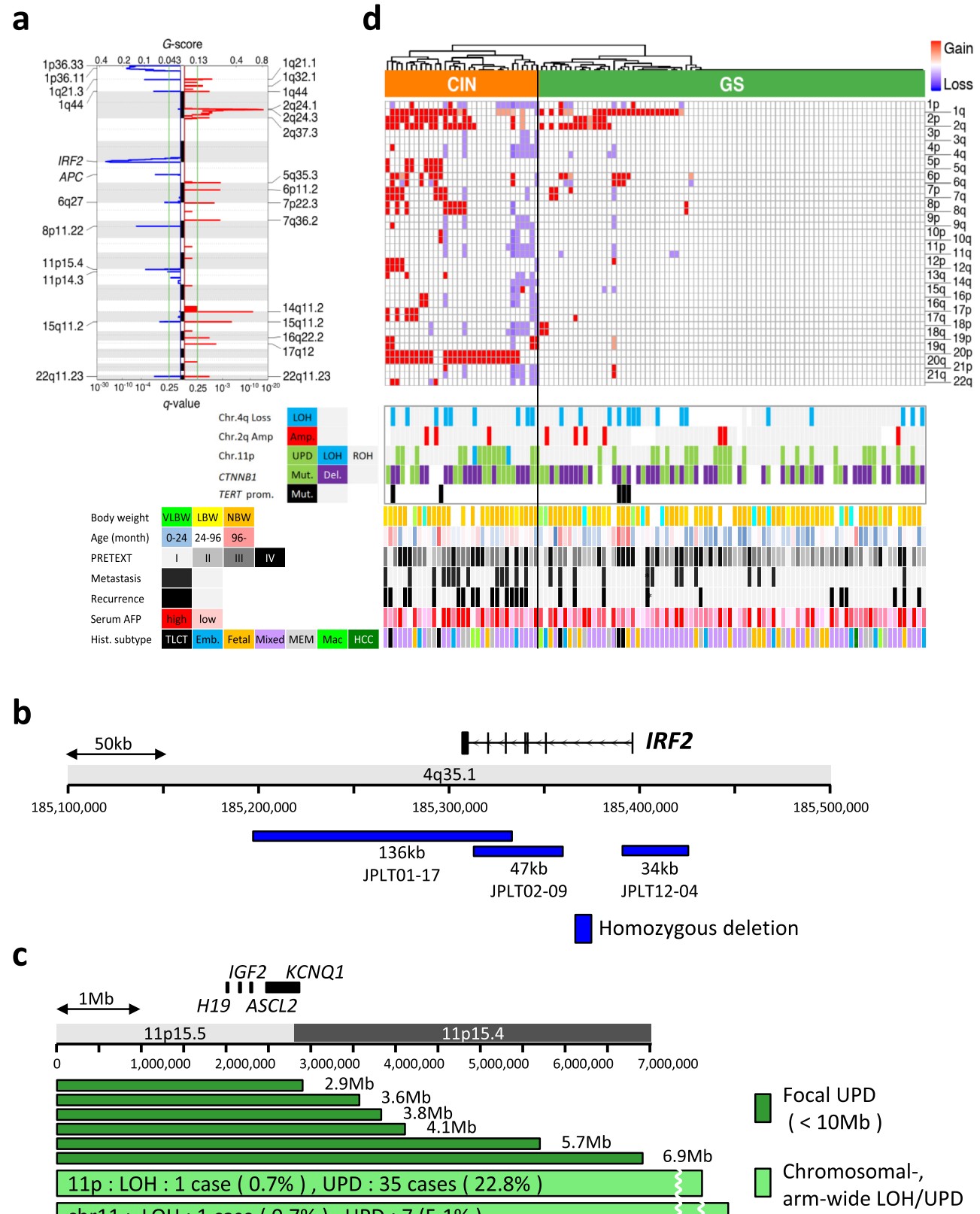

caused LOH of *IRF2* (Supplementary Fig. 5c). These cases exemplify the fragility of 4q in HB.

**Gene expression subtype in HB**. Following the paired-end RNA-sequencing for 111 HB and 28 non-cancerous liver tissues, consensus

clustering analysis identified three tumor subtypes and normal tissues: "proliferative" (46 cases), "mesenchymal" (24 cases), and "hepatocyte" (40 cases) (Supplementary Figs. 6, 7a). The "proliferative" HB subtype predominantly expresses cell cycle-related genes (*CCNB2, E2F1, MKI67, MYCN*) and *AFP*, suggesting its aggressive nature of cell proliferation. The tumors of this subtype also

**Fig. 2 Copy number aberrations and genomic stability of childhood hepatoblastoma (HB) genomes. a** Arm-wide chromosomal gain (left) and loss (right) were analyzed across 112 HB genomes using GISTIC 2.0. **b** Epicenter mapping of recurrent homozygous deletions in the vicinity of *IRF2* gene detected in 3 HB cases. **c** Epicenter mapping of focal LOH and UPD in chromosome 11p spanning *H19, IGF2, ASCL2,* and *KCNQ1* genes. **d**, A total of 112 HB tumors are hierarchically clustered into two groups based on arm-level somatic copy number alterations (SCNAs) defined by GISTIC 2.0; CIN (chromosomal instability, $n = 32$) and GS (genomically stable, $n = 80$). The heat map shows copy number statuses in each tumor (horizontal axis) plotted by chromosomal location (vertical axis) in the top panel. The middle panel represents the status of focal copy number (CN) changes in chromosomes 2q and 4p, LOH or UPD in chromosome 11p, and the driver mutations (*CTNNB1, TERT*). The bottom of the panel shows clinical and pathological parameters of HB patients. Statistical significance was analyzed by Fisher's exact test (*$p < 0.01$).

shared a high expression of the canonical Wnt-targeted genes (*LGR5, TBX3, BMP4, ASCL2, RNF43, DKK1, SP5, NOTUM,* and *GPC3*), which are compatible with a high frequency of *CTNNB1* mutation or deletions in this expression subtype. The gene sets upregulated in the "mesenchymal" tumors were characterized by the T cell receptors, matrix metalloproteinases (*MMPs*), and granzymes (*GZMs*), suggesting the intratumoral infiltration of cytotoxic immune cells (CTLs, NKT, and monocytes) with ECM degradation (Supplementary dataset 2). The "hepatocyte"-like tumors highly express the genes related to the metabolic functions of mature hepatocytes such as Cytochromes P450 (*CYPs*), UDP glucuronosyltransferases (*UGTs*), and metallothionein family genes (*MTs*). These results are consistent with those of the gene ontology enrichment analysis of the differentially expressed genes (Supplementary datasets 2 and 3).

We further examined the expression levels of HB-specific genes defined in the previously reported transcriptomic data[29]. As shown in the heatmap of Supplementary Fig. 7b, hepatic targets in the normal liver (*CYP1A1* and *CYP2E1*) were downregulated in most HB cases. Interestingly, negative regulators of Wnt-signaling (*SFRP1, SFRP5*) were upregulated in the "mesenchymal" HB but downregulated in the "hepatocyte" HB. Stemness-related Wnt targets were upregulated in the "proliferative" HB. Based on the 16 gene signatures[29], the C2 subtype of HB showed good overlap with the "proliferative" HB in our subtyping (Supplementary Fig. 7c).

**Epigenetic dysregulations in HB**. Genetic or epigenetic dysregulation of Chr.11p is the major feature of HB[9]. As shown in Supplementary Fig. 9, WGBS analysis revealed *H19* ICRs (maternally expressed) were hemimethylated in non-cancerous livers and HB with ROH (retention of homozygosity) but were fully methylated in HB with UPD, LOH, and LOI (loss of imprinting). *KCNQ1OT1* ICRs (paternally expressed) were unmethylated, mainly in UPD and LOH HB. For the large-scaled epigenomic profiling, we used Infinium HumanMethylation450 BeadChips (Illumina) for 146 tumors and 11 non-cancerous liver tissues. This BeadChip and WGBS showed a good correlation of methylation level in clinical samples ($R^2 = 0.951 \pm 0.004$). Methylation patterns of both known ICRs implicate preferential selection of paternal alleles during hepatoblast carcinogenesis. Human *IGF2* has at least five promoters regulated in a spatial and temporal manner[30,31]. *IGF2* Promoter 1 (Pr1), known as an adult human liver-specific promoter (Supplementary Fig. 10), was densely methylated in fetal liver (18 weeks of gestational age) but showed gradual hypomethylation in child HB livers in an age-dependent manner (Fig. 3). The hypomethylation was not correlated with the age at diagnosis and heterogenous in HB samples. On the contrary, Pr2 (fetal-tissue promoter) showed a gradual increase in methylation in non-cancerous liver samples and dense methylation in adult liver tissues. Although Pr3 and Pr4 promoters are also reported as fetus-specific and paternally imprinted, these CpG island promoters were constitutively unmethylated in our data. These methylation patterns of *IGF2* promoters implicate sustained epigenetic patterns of immature cells in the fetal liver.

We performed an unsupervised clustering analysis using promoter-based and enhancer-based probes (Supplementary Fig. 11). To define the promoter methylation epigenotype in 146 HB, we first performed promoter-oriented methylation analysis following the method used for the CpG island (CGI) methylator phenotype in TCGA studies. First, we removed the probes either designed on CH sites or located on Chr X and Y, and then extracted the probe near the transcription start sites within a distance less than 1500 bp. To focus on promoter hypermethylation, we removed the probes hypermethylated in normal liver tissues (>0.2 on average). Finally, we selected the most variable 5000 probes for the clustering analysis. Most probes were located within CGIs or CGI shores (Supplementary Fig. 12a). These loci were overlapped with active or bivalently marked promoters in embryonic stem cells or endoderm cells but became heterochromatin regions in HepG2 cells (Supplementary Fig. 12b). The promoter-based clustering analysis revealed the four promoter methylation subtypes (Supplementary Fig. 12c). The subtype P1 showed the most hypermethylated epigenotype and was closely related to the age at diagnosis ($p = 4.1 \times 10^{-15}$), high frequency of *TERT* promoter mutation ($p = 4.1 \times 10^{-4}$), TLCT subtype ($p = 5.3 \times 10^{-5}$), and chromosomal instability ($p = 2.6 \times 10^{-3}$). Twenty-two percent of the HBs showed an almost similar pattern to non-cancerous normal liver tissues (P4; "N-like"). Computational estimation of the immune cell fractions revealed the low values of tumor purity of this subtype. Although LOI in Chr.11p was mainly observed in the subtype P2 (moderately hypermethylated) and P1 ($p = 1.0 \times 10^{-3}$), UPD was mostly seen in P3 and P4 ($p = 0.0147$).

Next, we subtyped the same HB cohorts by the other set of probes located on the regulatory regions for hepatocytes (Fig. 4a). We extracted the probes overlapping with the hepatic enhancer regions defined by the H3K27ac chromatin immunoprecipitation (ChIP)-seq data of adult (ENCODE, GSM1112808, GSM1112809) and fetal livers (ENCODE GSM2343044), and then, selected the 1500 most variant probes. This enhancer-oriented clustering also defined four distinct subtypes. The subtype E1 tumors were well-overlapped with the P1 tumors of promoter-based subtyping, featuring the TLCT/HCN-NOS histology ($p = 2.5 \times 10^{-8}$) and *TERT* promoter mutation ($p = 3.0 \times 10^{-7}$). The subtype E2 showed a high frequency of copy number aberrations in the H19/IGF2 region (UPD and LOI, $p = 6.3 \times 10^{-4}$ and $1.8 \times 10^{-4}$, respectively) and proliferative pattern of expression signature ($p = 0.013$), suggesting embryonal phenotype with high mitotic activity. Upon comparison with the expression subtypes, the subtype "proliferative" was found significantly overlapping with the subtype E2 (Supplementary Fig. 13). The subtype E4 showed subtle changes in enhancer methylation, probably due to the low rate of tumor content ("Normal-like").

To predict the transcription factors that bind to the subtype-specific methylated regions, we performed motif enrichment analysis (Fig. 4b). Hypermethylation in the E1 showed enrichment of GC-rich sequences, implying hypermethylation in the vicinity of CpG-rich enhancers. Hypomethylation in the E4 showed enrichment of ETS factor recognition motifs, consistent with the intra-tumor infiltration of immune cells. Motif analysis of E2/E3-specific hypomethylation showed significant enrichment in DR1 sequences for hepatocyte regulators (HNF4, COUP-TFs,

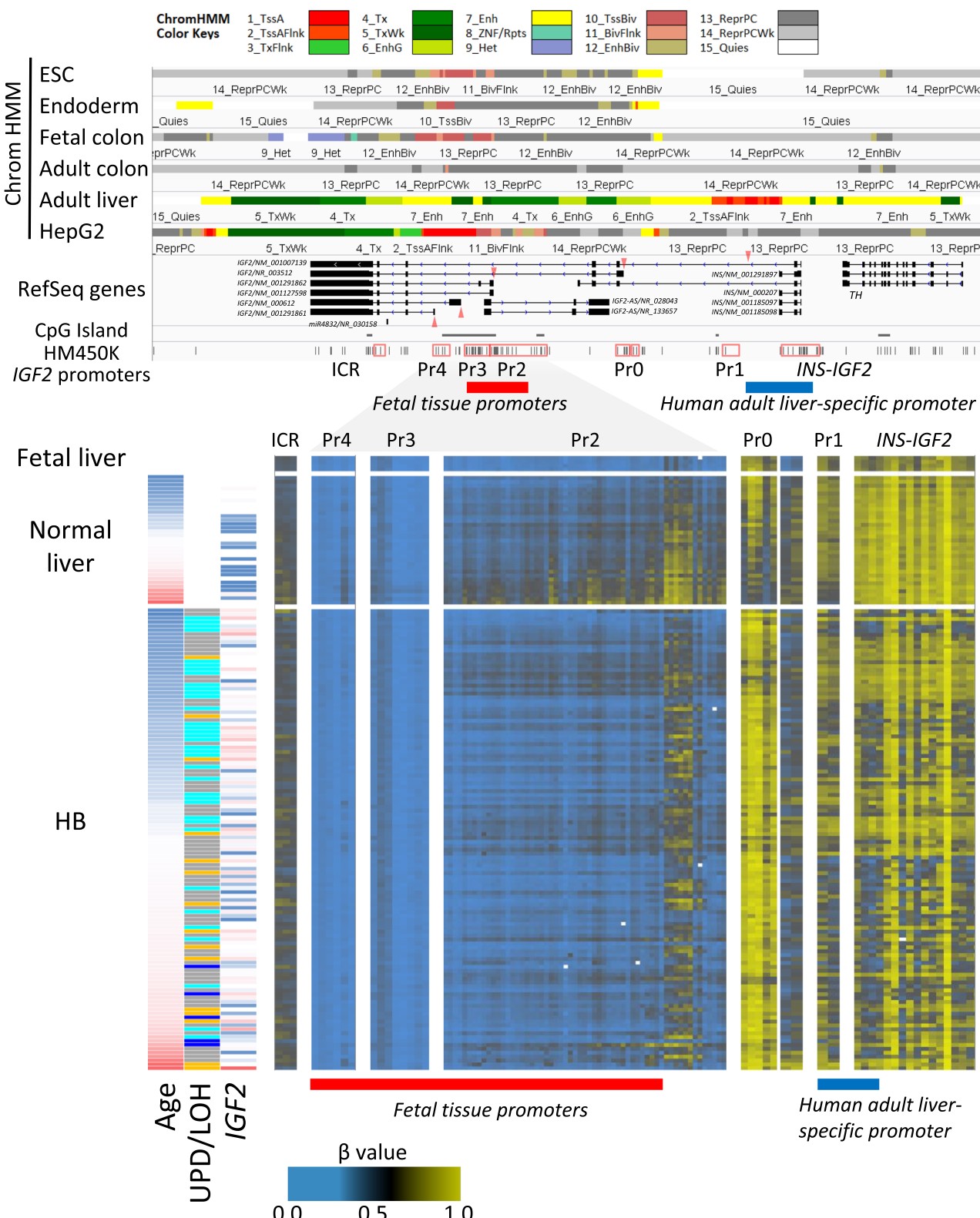

PPAR), suggesting that the hypomethylation of hepatocyte enhancers is sustained in the E2 and E3 HBs. Additionally, the E2-specific hypomethylation showed E-Box sequences (Myogenin and NeuroD1) as well as DR1 sequences. Among the several transcription factors recognizing E-Box elements[32], we focused on *ASCL2*, because of transcriptional upregulation in HB (Fig. 5a) and its essential role for the definitive endoderm and the tissue progenitor/stem cells in intestinal crypts[33].

**Regulatory transcription factor *ASCL2* in Wnt signaling.** In a recent report, strong enhancer regions upstream of *ASCL2* were identified by ChIP-seq analysis of the human intestinal crypts, where they are targeted by ASCL2 itself and CTNNB1[34]. Another report identified that the promoter of the Wnt-targeted long non-coding RNA, *WiNTRLINC1*[35], exists here. This long noncoding RNA (lncRNA) is reported to physically interact with the promoter region of *ASCL2* and enhance cis-acting regulation.

**Fig. 3 Methylation statuses of *IGF2* promoters in childhood hepatoblastoma (HB).** Methylation statuses of the *IGF2* promoters across 146 HB and 13 normal livers. The top panel represents chromatin statuses of human embryonic stem cells (H9, E003), their *vitro* derivative cells toward endoderm lineage (E011), fetal colon (E084), adult colon (E106), and adult liver tissues (E066), as defined in the Roadmap Epigenomics project (https://egg2.wustl.edu/roadmap/web_portal/index.html) around the *IGF2* gene (chr11:2,140,000-2,195,000). 1_TssA (Red), Active TSS; 2_TssAFlnk (Orange Red), Flanking Active TSS; 3_TxFlnk (LimeGreen), Transcr. at gene 5′ and 3′; 4_Tx (Green), Strong transcription; 5_TxWk (DarkGreen), Weak transcription; 6_EnhG (GreenYellow), Genic enhancers; 7_Enh (Yellow), Enhancers; 8_ZNF/Rpts (Medium Aquamarine), ZNF genes & repeats; 9_Het (Pale Turquoise), Heterochromatin; 10_TssBiv (IndianRed), Bivalent/Poised TSS; 11_BivFlnk (DarlSalmon), Flanking Bivalent TSS/Enh; 12_EnhBiv (DarkKhaki), Bivalent Enhancer; 13_ReprPC (Silver), Repressed PolyComb; 14_ReprPCWk (Gainsboro), Weak Repressed PolyComb; 15_Quies (White), Quiescent/Low. Red triangles represent the transcription start sites of five *IGF2* promoters. The heat map in the bottom panel shows methylation statuses of each tumor (vertical axis) plotted in order of the physical position along with the alternative promoters (horizontal axis). The left panel shows the age at diagnosis, aberration pattern (UPD, LOH, ROH and LOI) and the expression level of *IGF2* in each case.

Compared to the expression level of *ASCL2* in other adult cancers in TCGA Pan-Cancer transcriptome data (https://www.cbioportal.org/), the analysis showed that *ASCL2* was not expressed in adult HCC but highly expressed in colorectal and stomach adenocarcinoma (Fig. 5b). In addition, recurrent focal genomic amplification spanning *IGF2*, *WiNTRLINC1*, and *ASCL2* genes was detected in 9.9% of colorectal carcinomas and 2.7% of stomach adenocarcinomas (Fig. 5b, Supplementary Fig. 14). Immunohistochemical staining of ASCL2 showed high expression in HB as well as in neonate's intestine but no expression in normal liver, TLCT, or HCC. Upregulation of the transcription factor *ASCL2* raises the possibility that HB might have originated from the immature progenitor cell with a highly proliferative potential similar to the intestinal epithelial cells.

As *ASCL2* is located between the two well-known imprinting control regions (ICRs) of *H19/IGF2* and *KCNQ1/KCNQ1OT1* at chromosome 11p15, it shows mono-allelic expression (maternal) only in the placenta but the bi-allelic expression in other somatic tissues[36]. SNP typing on RNA sequencing data also revealed the monoallelic expression of these imprinted genes (Supplementary Fig. 15). Methylation analysis using WGBS and epigenotype microarrays revealed that the *WiNTRLINC1* promoter was densely methylated in normal child HB livers but hypomethylated in fetal livers and *CTNNB1*-mutated HB cases (Fig. 5e). *ASCL2* promoter is unmethylated in normal fetal and child livers but showed increased methylation in HB in an age-dependent manner (Supplementary Fig. 16). These promoter methylations of *ASCL2* and *WiNTRLINC1* were negatively correlated with the gene expression level of *ASCL2* (Supplementary Fig. 17). Therefore, hypomethylation of both regulatory regions might be necessary for the positive feedback loop to maintain a high expression level of *ASCL2*.

We further confirmed the specific binding of endogenous *ASCL2* by ChIP-sequencing of the HB cell lines, HepG2, and Huh6 (Fig. 5 and Supplementary Fig. 18). The specific binding sites of ASCL2 were well-overlapped with the active chromatin regions (H3K27ac- or H2K4me3-marked regions). They were unmethylated in fetal liver and HB, implicating that ASCL2 preferentially binds to unmethylated active chromatin regions. We also confirmed the co-localization of ASCL2 and CTNNB1 at several canonical Wnt-related genes. Regarding these co-binding regions, the strongly bound regions (ASCL2 ChIP score: >50) were stably unmethylated among non-cancerous liver tissues, but the moderately bound regions (ASCL2 ChIP score: 30–50) showed a gradual gain of methylation along with age (Supplementary Fig. 19). We further analyzed the differentially methylated regions ($n = 4683$) between fetal and adult liver tissues. Among the 3351 CpG sites hypermethylated in the adult livers, 53% showed sustained hypomethylation in HB, including *IGF2*, *ESR1*, *NR4A2*. On the contrary, among the 1332 CpG sites hypermethylated in the fetal liver, 19% showed sustained hypermethylation (Supplementary Fig. 20). Taken together, these results indicate that epigenetic dysregulation of *ASCL2* and *WiNTRLINC1*, as well as *fetal-liver-like*

*methylation patterns of IGF2 promoters,* may support the hypothesis of "cell of origin" in HB cells (Fig. 6).

**Molecular subtypes and clinical prognosis.** Figure 7 presents the relationship between the clinical outcome and the molecular parameters defined in this study. As shown in Fig. 7a, child HB (≥2 years of age), PRETEXT IV, distant metastasis (M), and other annotation factors (PVFENH) are significantly predictive for the 5-year event-free survival (EFS), which are consistent with the previous studies[4]. Concerning the molecularly defined subtypes (Fig. 7b), chromosomal instability ($p = 0.0083$, log-rank test) and methylation subtypes have great potential. The enhancer (E1, E2) and promoter subtypes (P1, P2) showed a significantly poorer outcome ($p = 0.0026$, 0.0005, respectively, by log-rank test). To identify a highly predictive surrogate marker, we visualized the methylation differences between tween and child HB (P1 + P2) and infant HB (P3 + P4) using a volcano plot (Fig. 7d). Hypermethylation of *DLX6-AS1*[37,38] (cg22421859, chr7:96622043 on hg19), the most significant classifier of these groups (cutoff: 0.4, Supplementary Fig. 21), demonstrated a high significance of 5-year EFS ($p = 0.0001$) and identified patients with a poor predicted prognosis, including HB cases of infant ($p < 0.0001$) or those with the positive annotation factor except for M ($p = 0.0232$) (Fig. 7c). Hypermethylation of the intronic region of *DLX6-AS1* is associated with its high expression in HB, which was also reported in several types of cancers[39]. As shown in the Sankey diagram (Fig. 7f), *DLX6-AS1* methylation might have the potential for risk stratification via enrichment of the methylation subtypes of poor prognosis (Fig. 7e).

**Discussion**

Although a recent assessment by the Children's Hepatic tumors International Collaboration (CHIC) suggested clinical and phenotypic diversity in HB[4], their molecular background has not yet been analyzed. Additionally, the number of patients included in previous studies was insufficient to determine inter-individual heterogeneity[40] or was applied only to a single molecular feature[29]. The comprehensive and integrative molecular analyses of primary childhood hepatic malignant tumors in our study reveal distinct diversity in genomic, epigenomic, and transcriptomic features in the large-scale JPLT-2 cohort[7].

First, whole-genome and whole-exome analysis revealed that the landscape of genomic aberrations in HB is closely associated with the age at diagnosis. Consistent with prior results, most HBs in this study were characterized by frequently mutated genes, *CTNNB1* and *TERT*. While somatic mutation/deletion of *CTNNB1* exon 3 was observed in more than 80% of HB patients younger than 8-years of age, promoter mutation of *TERT* was mostly seen in the tween HB cases, which suggests two distinct pathways of these childhood hepatic malignancies. These findings might indicate that the risk for an event increases with advancing

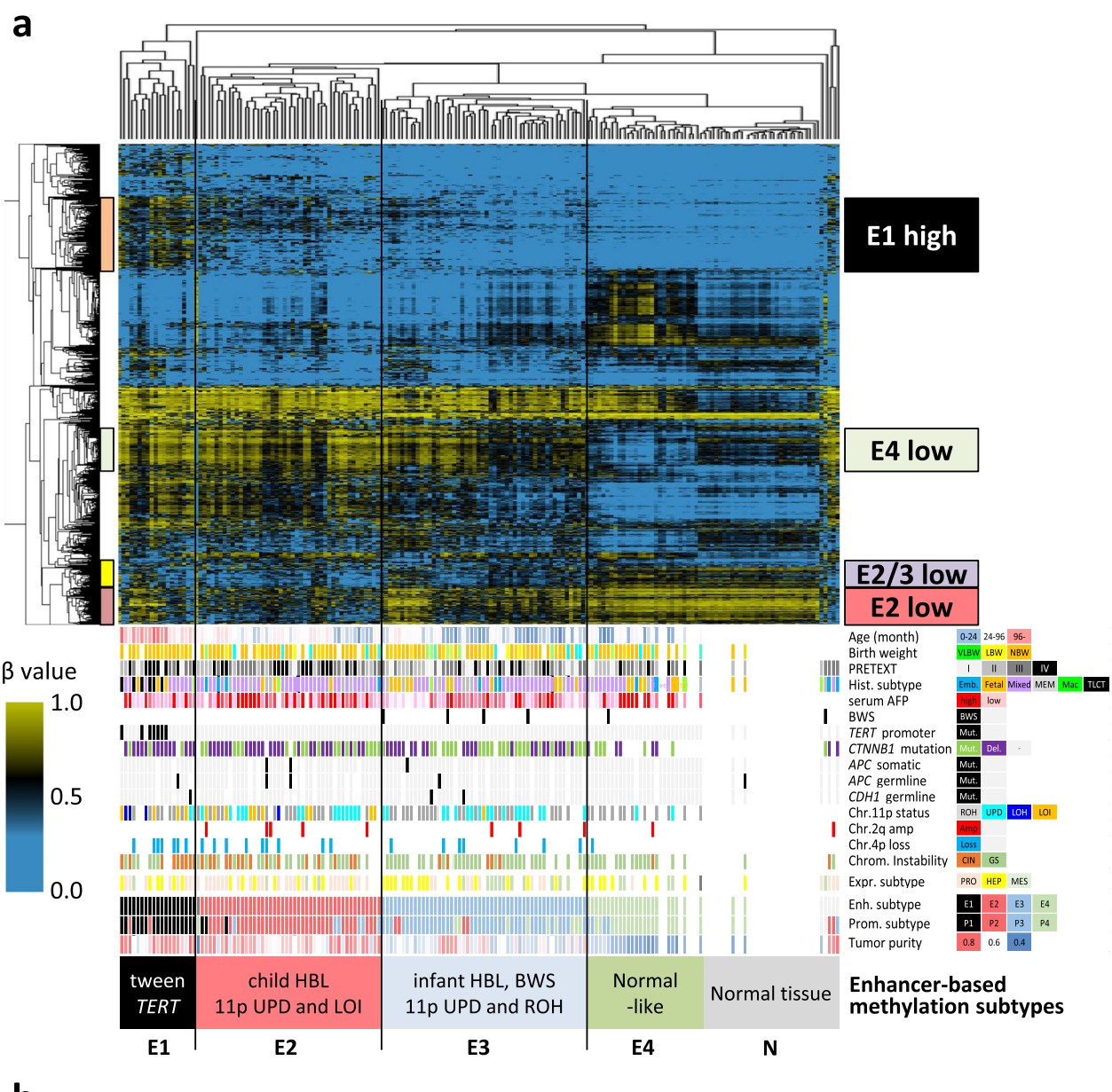

**Fig. 4 Enhancer methylation subtypes and aberrant hypomethylation in hepatoblastoma.** Unsupervised clustering of 146 childhood hepatoblastoma (HB) and 11 non-cancerous livers using the most variably methylated CpGs located at the enhancer regions of adult and fetal livers. The heat map shows the methylation level of each tumor (horizontal axis) plotted by the probe sets (vertical axis) in the top panel. The bottom panel shows clinical and pathological parameters, molecularly defined subtypes of HB patients. Statistical significance was analyzed using Fisher's exact test (*$p < 0.01$). **b** Significant motifs of transcription factors most strongly enriched in the subtype-specific hyper/hypomethylated probe sets using the TRAP (Transcription factor Affinity Prediction Web Tools). The matrix entries have an identifier that indicates one of the six groups of biological species (V$ vertebrates, I$ insects, P$ plants, F$ fungi, N$ nematodes, B$ bacteria). The *P*-value for each sequence was combined using the two-sided Fisher's method and corrected for multiple testing.

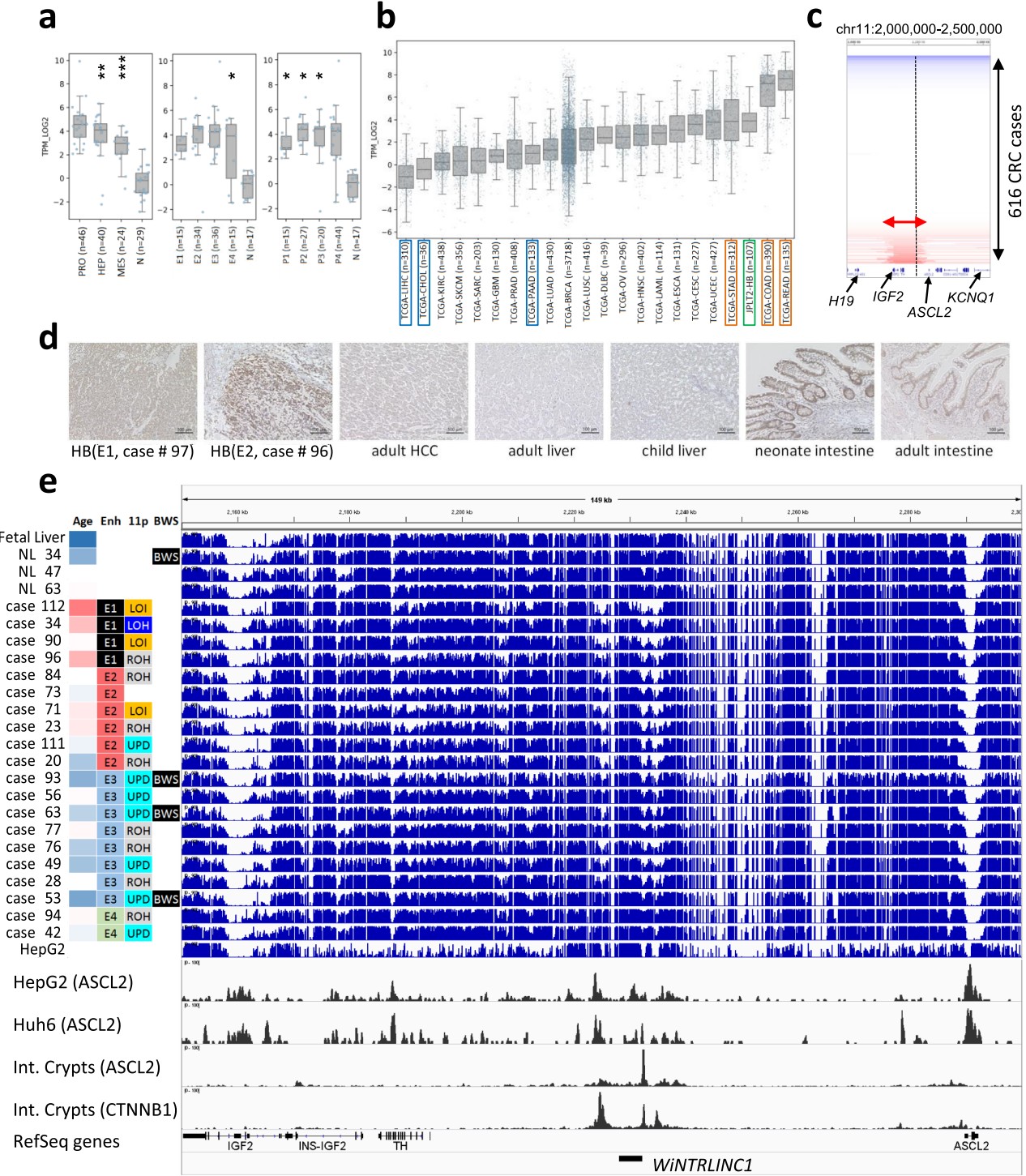

**Fig. 5 Aberrant hypomethylation in the *ASCL2* region. a** Expression level of *ASCL2* in hepatoblastoma (HB) in this study The center lines show the medians, the tops, and bottoms of boxes show quartiles, and the whiskers show the extremes within the range of the medians ± 1.5 × the interquartile ranges, wherein *$p < 0.05$, **$p < 0.01$, ***$p < 0.001$, ****$p < 0.0001$. Statistical significance was analyzed using a two-sided Fisher's exact test. Gene expression subgroup (left), PRO vs. N, $p = 0.221$; HEP vs. N, $p = 0.004$; MES vs. N, $p < 0.001$, Enhancer methylation subgroup (middle), E1 vs. N, $p = 0.229$; E2 vs. N, $p = 0.576$; E3 vs. N, $p = 0.351$; E4 vs. N, $p = 0.047$; Promoter methylation subgroup (right), P1 vs. N, $p = 0.013$; P2 vs. N, $p = 0.041$; P3 vs. N, $p = 0.025$; P4 vs. N, $p = 0.408$. **b** Expression level of *ASCL2* in hepatoblastoma (HB) in this study and adult cancers in TCGA Pan-Cancer project. **c** Recurrent focal amplification in *IGF2/ASCL2* region in TCGA colorectal carcinoma. **d** Immunohistochemical staining of ASCL2 using 100× diluted anti-ASCL2 rabbit polyclonal antibody (Biorbyt Ltd, Cambridge, UK), in the series of human liver, intestine, and tumor samples. This immunological analysis was repeated independently one other time with a similar result. **e** Landscape of the methylation statuses of the *IGF2/ASCL2* regions analyzed using whole-genome bisulfite sequencing (blue bar graphs, 21 HB, 3 normal livers, and the liver cancer cell line, HepG2). The bottom two rows (black) represent the ChIP-sequencing data (*CTNNB1* and *ASCL2*) obtained using the human intestinal crypts[35] and hepatoblastoma cell lines, HepG2 and Huh7. Clinical and molecular parameters (Age age at diagnosis, Enh enhancer methylation subtype, 11p copy number status of chromosome 11p, BWS Beckwith–Wiedemann syndrome) are shown on the left of the bar graph.

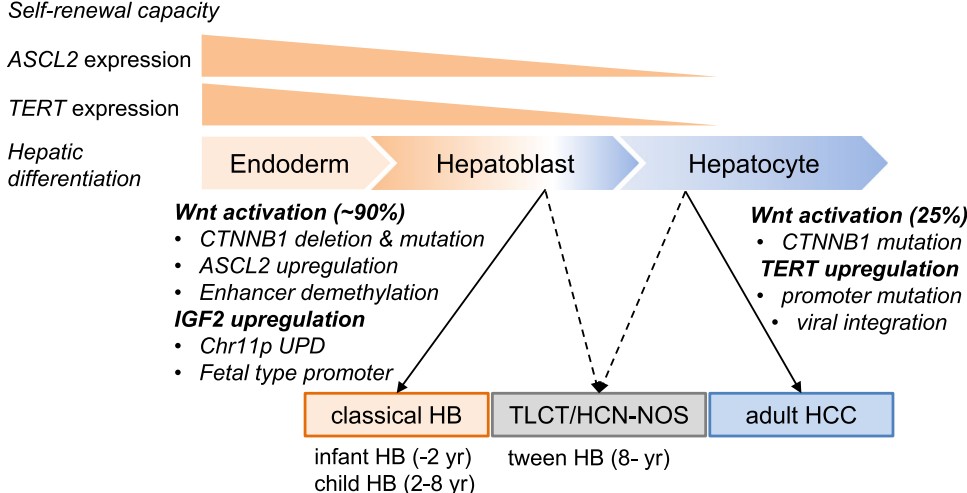

**Fig. 6 Overview of childhood and adult liver malignancies characterized by molecular aberrations and "cell of origin".** During the process of cell differentiation and maturation from definitive endoderm to hepatoblast and hepatocyte, the expression of ASCL2, a stem cell-related factor, and TERT, an immortality-related gene, is decreased, as well as the self-renewal capacity. Classical hepatoblastoma (HB), derived from hepatoblasts show high activation of Wnt and sustained upregulation of IGF2 due to genetic and epigenetic abnormalities, while adult hepatocellular carcinoma (HCC), derived from mature hepatocytes with low TERT expression, show its upregulation due to promoter mutations and viral integration. Wnt activation (25%) is mostly due to mutations in *CTNNB1*. TLCT/HCN-NOS shows combined features of both tumors. TERT: telomerase reverse transcriptase, TLCT: translational liver cell tumor, HCN-NOS: hepatocellular malignant neoplasm, not-otherwise-specified.

age at diagnosis of HB[8]. In addition, we also observed a positive correlation between the age at diagnosis and the number of SNVs. As the SNV burdens were remarkably few (fewest among the adult and childhood carcinomas), we further performed WGS of 33 HB to uncover the mutational spectrum. Based on the SNVs that we identified, the most contributing signatures are Signatures 1, 5, and 18. Signatures 1 and 5 are thought to have a clock-like property, whereas Signature 18 is a feature that often contributes to childhood malignancies such as neuroblastoma[41,42]. Although we compared these mutational patterns with the birth weight and the known prenatal risk factors of HB, we could not find any correlations between them.

Second, we performed copy number analyses, which revealed that 28.6% HB showed chromosomal instability while the remaining HBs were GS. Additional allele-specific calling of SNP arrays also revealed recurrent 11p15.5 UPD/LOHs, including the *IGF2/H19* region (31.6%), focal deletions in 4q35 (15.8%), and focal amplifications in 2q24.3 (7.0%) among GS tumors, which were predominantly in the child and infant HBs.

Third, we performed genome-wide methylation analysis with a focus on both CpG island promoters and hepatic enhancers. Motif enrichment analysis of the specific hypomethylated regions in the child and infant HBs suggests preferential binding of E-box transcription factors, such as ASCL2. This regulatory transcription factor plays an important role in maintaining Lgr5-positive intestinal stem cells[34]. While ASCL2 is crucial for definitive endoderm and digestive systems, it is not expressed during hepatocyte maturation. Ectopic overexpression of Ascl2 also occurs in murine intestinal neoplasia[43], and focal amplification or transcriptional upregulation of ASCL2 has been frequently observed not in human HCC but in human gastrointestinal cancer[44]. Thus, sharing the feedback upregulation of Wnt-targeted genes with colorectal cancer could explain the unique feature of Wnt activation in HB rather than in HCC, and may imply that their "cell of origin" is derived from the ASCL2-positive premature hepatoblast, similar to the intestinal epithelial cells, which have high proliferative potential. In our study, the expression level of *TERT* gradually decreased during hepatocyte differentiation, although high levels were sustained in the

gastrointestinal epithelial cells. Similar to this pattern, the classical HBs showed high TERT expression without promoter mutation. Alternatively, TLCT has a high frequency of *TERT* promoter mutations similar to adult HCC[18], which may function to avoid replicative senescence[45] and to maintain its self-renewal capacity during hepatocarcinogenesis (Fig. 6).

Lastly, we identified the molecularly defined subtypes in this large cohort of HB. In general, an examination of aberrant methylation is necessary for understanding the carcinogenic process of individual pediatric tumors. In the comparative analysis of primary and recurrent neuroblastoma, promoter methylation patterns were consistent over the course of the disease and were patient-specific[46]. In a previous study, comprehensive methylation profiling of 2801 tumors in the central nervous system (CNS) revealed 82 CNS tumor classes by machine learning and new CNS tumor entities[47]. In our analysis, promoter-oriented subtyping of HB was closely associated with the age at diagnosis, *TERT* promoter mutation, and copy number status of *H19/IGF2* regions. CIN was the significant parameter for poor prognosis, which is consistent with a previous report of Chr. 8q and Chr. 20 gain as a predictor of poor outcome[48]. Promoter methylation-based subtypes might also be predictive as an independent factor for survival of pediatric neoplasms[49,50]. In a previous report on RMS, we identified the novel methylation cluster related to the poor prognosis of embryonal RMS with PTEN silencing due to promoter hypermethylation[51]. Among the differential methylation of the defined epigenotypes, we identified a potential candidate for clinical application. The findings from our study indicate that the promoter hypermethylation subtype might be useful for risk stratification in addition to the clinical indicators. Additionally, hypermethylation of *DLX6-AS1* and *TERT* promoter mutation of TLCT (HCN-NOS) are promising as useful molecular markers for identifying recurrent risk factors in HB in future clinical trials. Although the functional role of *DLX6-AS1* during tumor progression has not been fully elucidated, its overexpression has been reported in several types of cancers (e.g., breast, lung, stomach, colon, liver) and is expected to be a potential therapeutic target[39].

In conclusion, this study describes a comprehensive molecular analysis of HB using members of the JPLT-2 cohort. Systematic molecular profiling of HB is essential to understanding the

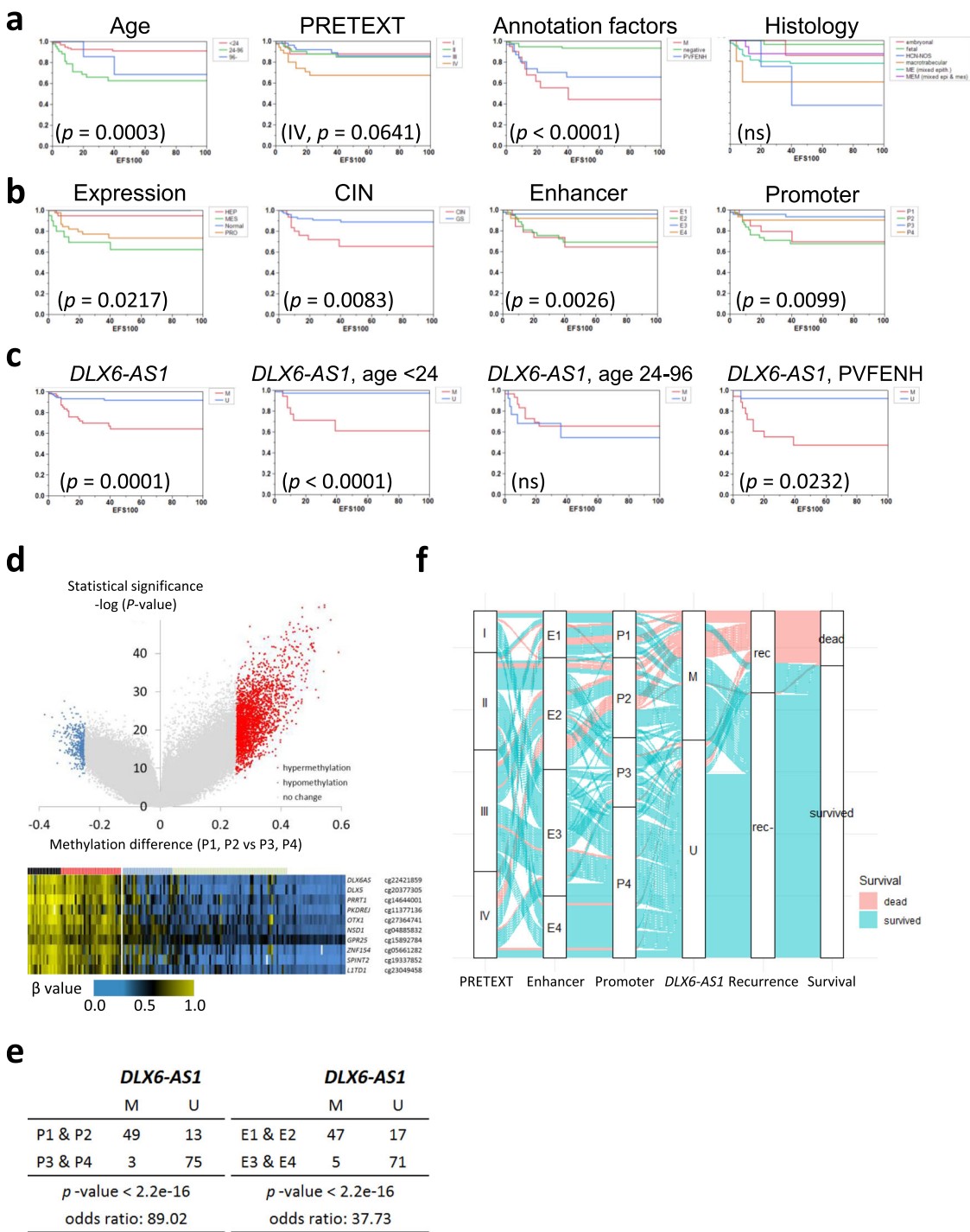

**Fig. 7 Prognostic significance of clinical parameters and molecularly defined subtypes. a, b** Samples were stratified according to the clinical parameters **a** and molecular-defined subtypes **b**. **a** Age, age at diagnosis; PRETEXT, the PRETEXT staging system; Annotation factors (M distant metastasis, + positive for VPERFNH, − negative for the annotation factors); Histology histologic subtypes. **b** Expression, gene expression; CIN, chromosomal instability; Enhancer, enhancer methylation; Promoter, promoter methylation. Statistical significance was analyzed using the log-rank test. **c** Surrogate methylation marker *DLX6-AS1* (cg22421859, chr7:96622043) for all childhood hepatoblastoma (HB) cases (left) and limited subgroup (middle, for infant HB; right, for AF-positive cases). **d** Volcano plots of differential methylation between older HB cases (P1, P2) and younger cases (P3, P4). The heat map in the bottom shows the methylation status of the 10 most significant CpG sites using the two-sided Fisher's exact test. **e** Surrogate methylation, *DLX6-AS1* to distinguish methylation subtypes. Statistical significance was analyzed using Fisher's exact test. **f** Sankey diagram stratifying the HB cases with poor prognosis by *DLX6-AS1* methylation.

epigenetic driver events that occur during hepatoblast carcinogenesis and provides clues that are necessary for risk stratification in precision medicine.

## Methods

**Patients and clinical information.** A total of 361 patients with HB (208 males and 153 females, median age 17.5 months) were treated during the JPLT-2 study between December 2000 and November 2012 at the institutions of the Japanese

Study Group for Pediatric Liver Tumors (JPLT)[7]. Two different protocols were used in this study: CITA (cisplatin and pirarubicin) as a first-line protocol and ITEC (ifosfamide, pirarubicin, etoposide, and carboplatin) as a second-line protocol. The Human Ethics Review Committee of Hiroshima University approved the study protocol. Signed informed consent for clinical data collection including age at diagnosis, collection and storage of biological samples, experimental analyses, and the publication of relevant findings was obtained from each parent (Approval of Ethics Committee No. Hi-78). The clinicopathological parameters and outcomes for these patients were analyzed during the JPLT-2 study. The clinical stages of disease were determined at the time of initial biopsy or resection according to the classification of the PRETEXT staging system, which was based on the number of liver segments involved, the extent of local invasion, the extent of regional lymph node involvement, and the presence of distant metastases.

The histology of HB was classified into two major subtypes: a well-differentiated (fetal) type and a poorly differentiated (embryonal) type in the JPLT-2 study that was reclassified by the international pediatric liver consensus classification[2]. Complete responses (CR) and partial responses (PR) of primary tumors by preoperative chemotherapy were evaluated using the Response Evaluation Criteria in Solid Tumors (RECIST) standards (http://www.recist.com/). The CR/PR cases were assigned as chemo-sensitive cases, and the stable disease/progressive disease cases were designated chemo-resistant cases.

**Cancer tissue sampling**. Tumor tissue specimens and their corresponding normal liver tissue specimens were obtained during surgery or biopsy from all patients who were analyzed before chemotherapy. The specimens were immediately frozen and stored at −80 °C until use. Tissues adjacent to the collected tissues were examined for diagnosis and confirmation by pathological testing.

**DNA and RNA extractions**. Tissue DNA samples were extracted by purification using standard phenol:chloroform methods. Total cellular RNA was extracted from tumor tissues by the acid–guanidinium–phenol–chloroform method[52]. The extracted RNA was quantified using a 2100 Bioanalyzer (Agilent Technology, Santa Clara, CA) with an Agilent RNA 6000 Nano kit. The RIN values of the extracted RNA were more than 7.4.

**Whole exome sequencing (WXS) and analysis**. Exome capture was carried out using SureSelect XT Exome V5 Custom kit (Agilent Technologies). The exome-captured libraries were sequenced on HiSeq2000/2500 with paired reads of 100–125 bp. One microgram of genomic DNA was fragmented using Covaris S2 (Covaris, MA, USA). Sequencing libraries were constructed using the TruSeq DNA LT Sample Prep Kit and TruSeq Exome Enrichment Kit (Illumina Inc., San Diego, CA, USA) according to the manufacturer's instructions. The average library size was 430 bp which corresponds to an average insert size of 300 bp that was assessed using the DNA High Sensitivity Kit on a 2100 Bioanalyzer (Agilent Technologies). Libraries were quantified by qPCR using a KAPA SYBR® FAST qPCR Master Mix (2X) Kit (Kapa Biosystems, Wilmington, MA) and 7900HT Fast Real-Time PCR System (Thermo Fisher Scientific, USA). Sequencing flow cells were prepared using a cBot (Illumina). NGS was performed on Illumina HiSeq 2500 (paired-end 101-bp runs).

**WGS and analysis**. For WGS, 500–600-bp insert libraries were prepared, according to the protocol provided by the TruSeq Nano DNA Sample Preparation kit (Illumina), and sequenced on HiSeq2000/2500 with paired reads of 100–125 bp (average depth: normal 29.0X, tumor 38.2X). Somatic SNVs and INDELs were called as previously described[23]. Briefly, WGS of matched tumor and normal were mapped onto GRCh37 using BWA and contrasted using our in-house software, Karkinos (https://github.com/genome-rcast/karkinos). The genotypes used in this study has been used in several studies[53–55], including the ICGC Liver cancer paper[18] and Clinical trials in the University of Tokyo Hospital[56]. The comparison between the different genotypes has been provided in Supplementary information[18] (ng.3126-S1.pdf), and a brief methodology has been described in the "Methods" sections of the previous papers. Somatic SVs were called using the GenomonSV software (https://github.com/Genomon-Project/GenomonSV).

For mutational signatures, we first tried de novo extraction by analyzing trinucleotide substitution patterns of 33 HB WGS using SignatureAnalyzer (https://software.broadinstitute.org/cancer/cga/msp). However, the optimal result consisted of a single mutational signature, indicating that the substitution patterns of 33 WGS were highly similar to each other. Therefore, we pooled the trinucleotide substitution patterns of 33 WGS and decomposed them into known mutational signatures from the Catalogue Of Somatic Mutations In Cancer (COSMIC) database (version 2) by using deconstructSigs[57]. The number of signatures used for signature decomposition was increased from 1 through 5, and the root mean squared error was monitored to find a plateau of decomposition accuracy.

**Somatic copy number alterations**. Genome-Wide Human SNP Array 6.0 (Thermo Fisher Scientific) was performed on 112 pairs of cancer and non-cancer DNAs, and their copy number alterations were analyzed using GEMCA[58]. Using GISTIC 2.0, CEL files were processed using Affymetrix Power Tools (v1.21.0) to extract log R ratios (LRR) and B-allele frequency (BAF). LRR and BAF of matched tumor–normal pairs were segmented using the R package ASCAT (v.2.5.2) with a segmentation penalty of 100. Copy number segments were uploaded to the public GenePattern server and analyzed using the GISTIC 2.0 module. For copy number clustering, hierarchical clustering was applied to arm-level copy number signals using the Euclidean distance and the Ward linkage algorithm.

**Germline variant calling**. Germline SNV/INDELs were detected using Genome Analysis Toolkit (v3.6). We applied base quality score recalibration to WES/WGS BAM files of normal tissues. Germline variants were called using HaplotypeCaller and were filtered by applying variant quality score recalibration. The remaining variants were further filtered using the following three criteria: first, we discarded common variants that had allele frequency >1% either in the Japanese population or in the 1000 Genome Project. Allele frequencies in the Japanese population were obtained from the human genetic variation database[59] and the integrative Japanese genome variation database[60]; second, we retained only variants that caused stop-gain or frameshift, that affected splice sites, or that were (likely) pathogenic according to the ClinVar database (as of March 5, 2019); third, we retained only variants that affected previously known cancer predisposition genes. We compiled and used a list of 40 genes whose mutations cause predisposition to tumors in various organs.

**Sanger sequencing of *CTNNB1* exon 3 and *TERT* promoter**. To detect a large deletion including exon 3 of the *CTNNB1* gene, genomic DNA samples derived from each tumor and the corresponding noncancerous tissues were amplified by PCR using a primer pair specific for exon 2 (5′-AAAATCCAGCGTGGACAAT GG-3′) and exon 4 (5′-TGTGGCAAGTTCTGCATCATC-3′) and analyzed using gel electrophoresis[61]. The breakpoint of exon3 deletions and point mutations in exon 3 was identified by capillary DNA sequencer ABI 3100 (Thermo Fisher Scientific). To detect the point mutations of *TERT* promoter lesion, the PCR products by the primers: 5′-CACCCGTCCTGCCCCTTCACCTT-3′ (TERT-2F) and 5′-GGCTTCCCACGTGCGCAGCAGGA-3′ (TERT-2R) were also analyzed by the capillary DNA sequencer. Mutation identification was confirmed with at least two amplification reactions from the original DNA.

**DNA methylation analysis**. The Illumina HumanMethylation450 BeadChip[62] was used to assay 146 HB and 11 non-cancerous liver samples. This platform included probes for more than 480,000 CpG sites, spanning 99% of RefSeq genes. Genomic DNA (500 ng) for each sample was treated with sodium bisulfite and recovered using the Zymo EZ DNA methylation kit (Zymo Research, Irvine, CA), according to the manufacturer's specifications. The raw signal intensity for methylated and unmethylated DNA was measured using a BeadArray Scanner (Illumina). After color-bias correction, background subtraction of the signal intensities, and inter-array normalization on Genome Studio (Illumina), the raw methylation value for each CpG was defined as $\beta = M/(M + U + 100)$, where $M$ and $U$ were the intensities of methylated and unmethylated probes, respectively. We then carried out a hierarchical clustering analysis using Cluster 3.0[63] with Euclidean distance and complete linkage. Computational estimation of non-tumor cell fractions was performed using InfiniumPurify software[64] with informative differentially methylated CpG sites for hepatocellular carcinoma (LIHC).

**Whole-genome bisulfite sequencing (WGBS)**. DNA from tissues was extracted using a phenol:chloroform method and the concentrations were measured using a Qubit® dsDNA BR Assay Kit on a Qubit® 3.0 Fluorometer (Thermo Fisher Scientific). DNA bisulfite conversion was performed by using EZ DNA Methylation-Gold (Zymo Research), according to the manufacturer's instructions, with DNA normalized to inputs of 500 ng from each sample. Sequencing libraries were constructed from 100 ng converted samples using the TruSeq DNA Methylation Kit (Illumina). The average library size is 630 bp which corresponds to an average insert size of 500 bp that was assessed using the DNA High Sensitivity Kit on a 2100 Bioanalyzer (Agilent Technologies). Libraries were quantified by Qubit® dsDNA HS Assay Kit on a Qubit® 3.0 Fluorometer. Sequencing was performed for 101 cycles using HiSeq Reagent Kit v2 on a HiSeq® 2500 instrument (Illumina, Inc., San Diego, CA, USA). Base-calling was done on the instrument and Fastq files were generated using bcl2fastq (Illumina). Using Bismark (v0.14.3) (Bowtie2 v2.2.5), WGBS reads were aligned to human genome (GRCh37/hg19) references, respectively, and methylation rates were calculated (average depth: 19.7X). Comparison analysis of WGBS and Infinium HumanMethylation450 BeadChip was performed at the CpG sites with the coverage depth of WGBS being more than 10.

**RNA sequencing**. RNA concentration was measured using a NanoDrop 2000c spectrophotometer (Thermo Fisher Scientific). RNA integrity was assessed using the Agilent RNA 6000 Nano kit on a 2100 Bioanalyzer (Agilent Technologies). The RIN data (the RNA integrity numbers higher than 0.6) was used to reflect the RNA quality for subsequent RNA sequencing. Sequencing libraries were generated using the TruSeq RNA LT Sample Prep Kit (Illumina), according to the manufacturer's instructions, with RNA normalized to inputs of 1 μg total RNA from each sample. Sequencing was performed for 101 cycles using HiSeq Reagent Kit v3 on a HiSeq® 2500 instrument (Illumina). Base-calling was performed on the instrument and Fastq files were generated using CASAVA (Illumina) with default settings for the RNA-seq data. Fastq files of RNA-seq reads were mapped to the human genome

(GRCh37/hg19) references, and human transcriptome database (UCSC gene), respectively, in order to map splicing reads and unspliced reads accurately. The Burrows–Wheeler Aligner was used as mapping software. After the transcript coordinate was converted to genomic positions, reads having smaller mismatches are chosen either from the transcript or genome mapping result. Further local realignment was performed using an in-house short reads aligner with a smaller $k$-mer size ($k = 11$) to resolve the alignment near indels and small exons. Finally, fragments per kilobase of exon per million fragments mapped (FPKM) values were calculated for each UCSC gene entry by using reads covering genomic positions considering strand-specific information.

For the clustering analysis, we selected the 5000 most variable genes from our data. After the normalization and scaling of the FPKM values, we applied consensus clustering with the Euclidean distance by the $k$-means algorithm with 1000 repetition sampling 80% of individuals in each run, for $k = 2–8$ using the R package[65], ConsensusClusterPlus (https://bioconductor.org/packages/release/bioc/html/ConsensusClusterPlus.html). The choice of the number of clusters was made based on the consensus matrices for each $k$, the area under the empirical cumulative distribution curve, delta area, and tracking plot. Gene ontology analysis of the upregulated and downregulated gene sets in each subgroup was performed using DAVID Bioinformatics Resources 6.8 (https://david.ncifcrf.gov/home.jsp).

**Motif-enrichment analysis for differentially methylated regions**. We selected the probe sets of "E1 high ($n = 136$)," "E2 low ($n = 157$)," "E2/3 low ($n = 55$)," and "E4 low ($n = 84$)" for the motif analysis to predict the binding of specific transcription factors. We extracted the sequence around the probes (800 bp) and analyzed the enrichment of TF recognition motifs using Transcription factor Affinity Prediction (TRAP) Web Tools (http://trap.molgen.mpg.de/cgi-bin/home.cgi)[66]. We used the multiple sequence mode with the reference of the "transfac_2010/1 all_matrices" and the background model of human promoters. Multiple test correction was performed using the Benjamini–Hochberg method.

**ChIP-sequencing**. HB cell lines, HepG2, and Huh6 were obtained from the Cell Resource Center for Biomedical Research at Tohoku University (Sendai, Japan) and the Japanese Collection of Research Bioresources (Osaka, Japan), respectively. The cells were cultured in Dulbecco's modified Eagle medium (DMEM) (Life Technologies) supplemented with 10% fetal bovine serum (FBS) and penicillin/streptomycin. ChIP using anti-ASCL2 mouse monoclonal antibody (MERCK, MAB4418, 1:400 dilution) and anti-CTNNB1 rabbit polyclonal antibody (Santa Cruz, sc-7963, 1:200 dilution) was performed as previously reported. Briefly, the cells were cross-linked with 1% formaldehyde for 10 min at room temperature, and cross-linked cell lysates underwent ultrasonic fragmentation and were incubated with antibodies bound to Dynabeads (Thermo Fisher, #10001D and #10003D) at 4 °C overnight. The beads were washed and eluted with elution buffer (0.5% SDS, 25 mM Tris–HCl, 5 mM EDTA). The eluates were treated with 1.5 μg of pronase at 42 °C for 2 h, then incubated at 65 °C overnight to reverse the cross-links. The immunoprecipitated DNA was purified by QIAquick PCR Purification Kit (QIAGEN, #28106). Sequencing libraries were generated using the TruSeq ChIP Sample Prep Kit (Illumina), according to the manufacturer's instructions.

Sequencing was performed for 50 cycles using HiSeq Reagent Kit v3 on a HiSeq® 2500 instrument (Illumina). Reads were mapped using bowtie-1.2.1.1 and samtools-0.1.16. Local coverage was calculated by the deepTools package (bamCoverage 2.5.4) with a smoothing length of 300 bp (smooth length = 300). The coverage of an alignment of reads or fragments (BAM file) is calculated as the number of reads per bin, where bins are short consecutive counting windows of a defined size. A coverage track was generated as a bigwig file. *bamCoverage* offers normalization by scaling factor, reads per kilobase per million mapped reads (RPKM).

**Immunohistochemistry**

*Antibodies*. An affinity-purified polyclonal rabbit antibody against ASCL2 was raised against a 61–110 amino acid peptide sequence that was mapped in the middle of ASCL2 (No. ORB155740, Biorbyt Ltd, Cambridge, UK) and used at a concentration of 2.5 μg/ml.

*Tissue preparation*. The tissues were cut into 4-μm-thick serial sections. Sections were deparaffinized and rehydrated through ascending grades of alcohol to phosphate-buffered saline (PBS) at pH 7.4. Heat-based antigen retrieval was performed as follows: sections were treated for 15 min in 0.01 M citric acid buffer, pH 6.0, under 2 atm, and at 121 °C using an autoclave. After decreasing the pressure, sections were removed and permitted to cool for ~20 min before being washed thrice in PBS for 5 min. Endogenous peroxidase was quenched in 0.3% $H_2O_2$. After washing twice with PBS for 5 min, nonspecific antibody binding was blocked by incubating the sections in protein blocking solution (Dako, Carpinteria, CA) for 10 min. Sections were then transferred to a humidified chamber and incubated in 100× diluted antibody solution overnight. Following this and subsequent incubations, the sections were thoroughly washed three times with PBS for 5 min each. For ASCL2 immunohistochemical staining, the sections were incubated in the labeled streptavidin-biotin polymer (Envision Plus, Dako), followed by the addition of 0.05% 3,3′-diaminobenzidine in distilled water with $H_2O_2$ as a substrate. Sections were lightly counterstained with Mayer's hematoxylin and then mounted.

**Statistics and reproducibility**. Unless specified otherwise, Fisher's exact test was used for $p$-value calculations between two categorical variables. Wilcoxon rank-sum and Kruskal–Wallis tests were used to analyze differences between two or more than two continuous variables. Log-rank tests were used to compare survival distributions in Kaplan–Meier plots. FDR corrections were performed for multiple testing corrections of gene ontology analysis by using the linear step-up method of Benjamini and Hochberg.

**Reporting summary**. Further information on research design is available in the Nature Research Reporting Summary linked to this article.

## Data availability
The expression level of *ASCL2* in adult cancers in TCGA Pan-Cancer transcriptome data is available at the GDC portal site https://portal.gdc.cancer.gov [https://portal.gdc.cancer.gov/query?filters0=%7B%22op%22%3A%22and%22%2C%22content%22%3A%5B%7B%22op%22%3A%22and%22%2C%22content%22%3A%5B%7B%22op%22%3A%22in%22%2C%22content%22%3A%7B%22field%22%3A%22cases.project.program.name%22%2C%22value%22%3A%5B%22TCGA%22%5D%7D%7D%2C%7B%22op%22%3A%22and%22%2C%22content%22%3A%5B%7B%22op%22%3A%22in%22%2C%22content%22%3A%7B%22field%22%3A%22cases.samples.sample_type_id%22%2C%22value%22%3A%5B%2201%22%2C%2202%22%2C%2203%22%2C%2204%22%2C%2205%22%2C%2206%22%2C%2207%22%2C%2208%22%2C%2209%22%5D%7D%7D%7D%2C%7B%22op%22%3A%22in%22%2C%22content%22%3A%7B%22field%22%3A%22files.analysis.workflow_type%22%2C%22value%22%3A%5B%22HTSeq%20-%20Counts%22%5D%7D%7D%5D%7D%5D%7D&query=cases.project.program.name%20in%20%5B%22TCGA%22%5D%20and%20%20cases.samples.sample_type_id%20in%20%5B%2201%22%2C%20%2202%22%2C%20%2203%22%2C%2204%22%2C%2205%22%2C%2206%22%2C%2207%22%2C%2208%22%2C%2209%22%5D%20and%20files.analysis.workflow_type%20in%20%5B%22HTSeq%20-%20Counts%22%5D]. The H3K27ac chromatin immunoprecipitation (ChIP)-seq data of adult (GSE96504) and fetal livers (GSM1598036) in ENCODE Roadmap Project. The raw and processed sequence data of WGS are available under restricted access, access can be obtained by contacting the National Bioscience Database Center (NBDC) Human Database (hum0161). The raw and processed sequence data of WXS (Exome), RNA-seq, WGBS and microarray data (SNP array and methylation array) are available under restricted access, access can be obtained by contacting the NBDC Human Database (hum0233). The ChIP-sequencing data generated in this study are publicly available in the Gene Expression Omnibus (GEO) under accession number GSE169566. The remaining data are available within the Article, Supplementary Information or Source Data file. Source data are provided with this paper.

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

## Acknowledgements

We gratefully acknowledge the technical staff in RIKEN IMS for library preparation and sequencing, and Hiroko Meguro for microarray experiments. The super-computing resource 'SHIROKANE' was provided by the Human Genome Center, The University of Tokyo (http://supcom.hgc.jp/). We are grateful to the donors and donor families for granting access to the tissue samples. Funding: This study was supported by AMED

(P-DIRECT, JP18CK0106332, JP18LK0201066 and JP19cm0106502), MEXT (Grants-in-aid for Scientific Research-A No. 15H02567 and 16H02778), and BioBank Japan.

## Author contributions

G.N., H.A., E.H., H.N. designed the experiments. E.H., S.K., M.K., T.H., K.W., K.I., M.Y., Y.T., T.I. performed tissue sampling and library preparation. G.N., S.Y., M.F., T.F. T.U. analyzed the data. Y.H., A.N., A.H., E.H. performed validation experiments. G.N., M.F., H.A., E.H., H.N. wrote the manuscript. All authors read and accepted the manuscript.

## Competing interests

The authors declare no competing interests.
