## [Peer Review File · Nature Communications]

REVIEWER COMMENTS

Reviewer #1, expert in liver genomics (Remarks to the Author):

In this work, Nagae et al. conduct an integrated characterization of pediatric liver cancers (mostly hepatoblastoma) with a combination of genomic (whole exome/genome sequencing), transcriptomic (RNA-seq) and epigenomic (methylation arrays and WGBS) methods. Using these valuable data sets, they provide a comprehensive description of driver events, mutational processes and molecular subgroups in hepatoblastoma, with a strong focus on DNA methylation changes.

This is overall a well conducted study that will provide important data sets and results to the community. The global genomic characterization is convincing, as well as the description of DNA methylation-based subgroups. However, the focus on ASXL2 transcription factor as a major regulator of hepatoblastoma suggesting a "cell of origin" for these cancers is enigmatic and not justified by the presented data in my opinion (see question 12 below). I also have many questions and concerns regarding the Methods, that are often described too briefly to allow reproducibility of the results.

1) The authors should provide a detailed clinical annotation with details for each sample, not just a summary at the cohort scale as in current Sup Table 1. In addition, given that each omic analysis is performed in only a subset of the samples, the authors should indicate what analysis was performed for each sample.

2) Pediatric HCC usually develop in a context of constitutional liver disease (tyrosinemia, mitochondrial cytopathies...) and often liver cirrhosis. What were the etiologies of the 9 cases in this cohort, and did the authors identify causal germline variants in HCC?

3) The Methods section related to variant calling is very brief. In particular the tools for alignment and variant calling from whole exome sequencing data are not mentioned. No reference is provided regarding the GenomonSV tool used for SV calling, so I suppose it has not been published yet? If so, the authors should provide minimal description of how it works and benchmark against peer-reviewed alternative tools.

4) The authors state that they identified "26 significantly mutated genes". However, they do not explain how recurrence significance was assessed. Did they account for gene size and genomic covariates using established methods like MutSigCV? Or did they look for an enrichment of functionally important mutations? This is an important point as most candidate driver genes in Fig. 1a are mutated in just 3 samples (2.2%) and they contain large genes like TTN, known to be recurrently mutated in any data set because of its size.

5) The authors used a list of 40 cancer predisposition genes to identify putative causal germline variants. However, they do not provide this list nor explain how it was compiled. In addition, since all mutated genes are tumor suppressors, it would be informative to look for somatic second hits (mutation or loss of the wild-type alleles) in matched tumors. In particular, do the 3 patients with focal 5q22.2 deletions harbor germline APC mutations?

6) The authors used deconstructSigs to decompose substitution patterns into known COSMIC mutational signatures. However, they do not indicate what pool of COSMIC signatures was used. Did they consider the complete panel of 30 signatures in version 2 or 49 signatures in version 3? How did they end up with 3 signatures? Were other signatures not present at all in tumors or did they apply a minimal threshold to consider a signature as operative?

7) Four subtypes are defined based on transcriptomic profiles. However, no details are provided regarding how the clustering was obtained (gene selection, distance metric, clustering method...), precluding reproducibility of the classification. Also, the authors should assess the stability of the clusters (e.g. by consensus clustering) and explain why they chose the number of 4 clusters. Importantly, several classifications of HB were previously described, in particular the C1/C2 related to prognosis (Cairo et al., Cancer Cell 2008). The authors should compare their classification with

previous work to determine if their groups match previously defined subtypes or define new molecular entities. Finally, it is unclear to me why the third group is called "Mesenchymal", although it mostly shows an overexpression of immune genes?

8) In Sup. Fig. 7, does the heatmap (bottom panel) correspond to methylation levels assessed by microarray? It is not clear from the legend. Same question for Fig. 3. Also no legend is provided in Fig. 3 regarding age, UPD/LOH and IGF2 annotations. The analysis of IGF2 promoters is interesting but the conclusion is a bit enigmatic to me. In Fig. 3, we can see that Pr1 (adult promoter) becomes unmethylated and Pr2 (fetal promoter) becomes methylated with age in normal liver, as expected. Some HB (mostly older) display Pr1 demethylation, whereas the methylation level of Pr2 is heterogeneous. Sup. Fig. 7 shows detailed methylation patterns of the promoters with WGBS. A fetal liver sample (of what age?) is shown as reference but no adult liver. Overall, it is difficult to conclude from these data that HB keep a fetal promoter usage for IGF2.

9) According to the main text, Sup. Fig. 9 represents a classification based on the 5,000 most variant probes located in CpG islands, whereas the figure legends mentions a classification based on promoter regions. How were these CpG sites selected? Indeed, only a fraction of promoters contain CpG islands, and a large proportion of CpG islands are not located at promoters...

10) Could the authors comment on the overlap between gene expression and DNA methylation-based classifications? It seems to me that they are not strongly correlated, except maybe the "normal-like" groups? How do the authors interpret this lack of association?

11) The Methods section for motif analysis of differentially methylated enhancers is missing. What regions were considered and what did they use as genomic background? Fig. 4b is hard to understand. What does "Matrix_name" stand for and the "V\$" or "F\$" before transcription factor names? The authors mention a hypermethylation of SP1 motifs in E1 but I do not see SP1 among the top transcription factors in Fig. 4b.

12) I do not understand why the authors dedicate a specific focus to ASCL2 transcription factor. It does not appear in the top of the list in Fig. 4b, and I suppose there are many transcription factors upregulated in hepatoblastoma. I do not see why the parallel with intestinal crypts would be relevant for hepatoblastoma. ASCL2 seems to be outside the minimal region of gain in Fig. 5c. The authors do observe methylation changes in ASCL2 region but I do not see how they can causally link these changes with hepatoblastoma pathogenesis, nor why they link them with "classical HB" in the abstract. Overall, this part seems largely correlative and speculative to me.

13) The authors identify several clinical and molecular features associated with poor event-free survival. However, significant molecular markers (CIN and methylation groups) are also related to age. Are these markers significantly associated with EFS independently from age? Regarding methylation of DLX6AS, what beta-value threshold was applied to classify tumors as "M" or "U", and how was it chosen? DLXAS methylation status (Fig. 6c) seems to largely reproduce the age classification (Fig. 6a). Don't we get the same curve by simply classifying patient below / above 2 years old? Also, DLXAS seems largely unmethylated in infants (P3, P4) according to Fig. 6d. So how did the authors define an "M" and "U" group among infants. Did they modify the beta value threshold, and how?

14) In the Discussion, the authors state that mutational signature 18 is "unique to childhood malignancies". This is not true as this signature was found in many adult cancers (Alexandrov et al., Nature 2020).

Reviewer #2, expert in methylation and epigenomics (Remarks to the Author):

This is a comprehensive molecular study that integrated genomic, transcriptome and epigenetic profile of a large cohort of hepatoblastoma. Their findings were largely consistent with previously published genomic studies. They showed an overall low mutation rate with CTNNB1 and TERT being the most commonly mutated genes, and frequent dysregulation of chromosome 11p. They

also identified molecular subtypes stratified by gene expression profiles and epigenetic profiles and their association with clinical outcome of the patients. The study is well designed and the data is well presented.

The finding of the upregulation of E-box recognizing transcription factor ASCL2 is potentially interesting. However, detail functional studies on the potential role of ASCL2 in hepatoblastoma and the interaction with Wnt signaling are lacking. Thus, the functional studies of ASCL2 is strongly advised to support the functional importance of ASCL2.

The author identified hypermethylation of DLX6AS as single most significant prognostic classifier associated with poorer prognosis. Is this purely as an association or has functional significance? Would the hypermethylation of DLX6AS associated with down-regulation in hepatoblastoma? This should be demonstrated. It has been reported in previous studies that long noncoding RNA DLX6AS promoted tumorigenesis and cancer progression in several cancer types, e.g breast cancer and esophageal cancer. The authors shall address the discrepancy to previous findings.

A number of large scale genomic studies on hepatoblastoma have been published previously. The authors should include a comparison of their finding with published data in the discussion. For example, NFE2L2 is found to be frequently mutated in other studies but seems no mutation was detected in the current study.

Reviewer #3, expert in epigenomics (Remarks to the Author):

Nagae et al. performed a deep genetic, transcriptional and epigenetic analysis of 163 (154 hepatoblastomas and 9 hepatocellular carcinoma) samples. They gather and interpret an impressive number of different genetic, transcriptional and epigenetic. They use these comprehensive data for an integrated analysis combining the diversity in the genetic background (driver, somatic and germline mutations), a classification according to tumor specific expression profiles (4 major groups) and DNA-methylation based grouping based on enhancer signatures. They finally extend their analysis towards a specific set of tumors with LOH/LOI on 11p i.e. subsets of HB with imprinted deregulation. Finally they combine their molecular groupings to stratify the relative survival in relation to genetic and epigenetic data.

The impressive data provide a deep insight in pediatric HB (and HCC) and represent a very rich and unique resource of pediatric liver cancers. In their first chapter describing the "landscape of germline and somatic mutations" the authors find a relatively low number of somatic and germ line mutations most of them occurring in known WNT and TERT related genes including APC (germline). These mutational signatures were neither correlated to age nor birth weight. Using arrays and sequencing they further identified a series of frequent structural variants, copy number alterations and signatures of genome instability – these classifications are later of relevance. A following transcriptional classification into four major subtypes identifies mesenchymal, proliferative, normal and classical signatures each with a relative dominant expression of either cell cycle regulators, WNT signaling genes, immune-specific signatures or normal hepatospecific profiles, while hepatic genes are generally downregulated in all four subtypes.

Finally the authors screen for DNA-methylation changes using 450K arrays and for a subset of 21 cases with either LOH or LOI on chromosomes 11p using WGBS. To identify DNA methylation dependent subclusters they focus on promoters and enhancers (identified in ENCODE datasets). They identify 4 Promoter and 4 Enhancer "classes" and subsequently focus their interest of the enhancer classes to concentrate on binding sites for ASCL2. Their analyse of the WGBS data identifies hypomethylation signatures in certain fetal IGF2 promoter(s) concomitant with a ASCL2 upregulation. They claim that this supports the hypothesis that a distinct class of HB develops from fetal (i.e. intestinal epithelial cell) progenitors.

Finally they use the classified molecular signatures to stratify molecular changes in HB and identify chromosomal instabilities and methylation signatures as major predictors for the outcome. They show predictive signatures in 10 genes with DLX6AS as a most prominent example.

My major criticism of the paper is that text is difficult to read and that the main messages are not well worked out. Furthermore the authors spike the text with (too) many abbreviations. In the methylome analysis I do not understand the exclusive focus in enhancer Signatures. The authors

should show a comprehensive 450K analysis also to rationalize why they focus on enhancers. Furthermore it's unclear why they do not apply tools to estimate the immune cell type contributions as e.g. the LUMP assay (integrated in RnBeads and other packages) or apply deconvolution tools to define covariant CpGs.

Major comments:

1) The main genetic conclusions are extremely difficult to extract for the reader. Some descriptions are rather detailed and the text is full of abbreviations, brackets etc. disturbing the reading flow and blurring main messages. This part requires a much more clear structure.

2) Gene expression:

- line 219: how were the subtypes annotated (classical, mesenchymal etc.?) or are they simply termed like that? This should be described more clearly.

- here a differential expression analysis between the subtypes would be interesting to find differential genes with statistical significance

3) Epigenetics:

- before describing the enhancer specific focus and locus-specific findings (e.g. H19) it would be interesting to obtain information about overall/genome-wide findings from WGBS and the Infinium analysis. How do 450K and WGBS data correlate?

- how were the described loci/genes selected? In the method section I only found a note on hierarchical clustering. A differential methylation analysis between groups or regression analysis in relation to age would make sense to get a genome-wide impression and not just glimpses at individual sites. The volcano plot in Figure 6D indicates that a differential methylation analysis between different ages was already done. However, I cannot find it in the method section and it should be described more clearly in the results part.

- to me the finding that methylation patterns at the IGF2 promoter resembles immature fetal cells on its own is not conclusive enough as it is a single locus. How similar are methylation patterns of HB and fetal liver cells on a genome-wide scale? The 'memory' of a fetal hepatoblast origin should be detectable in more regions.

4) ASCL2

- line 302: is this upregulation also detected in the HB RNA-seq data?

- Is it known whether ASCL2 preferentially binds to unmethylated/methylated binding sites? How is the methylation status at ASCL2 binding sites in HB and fetal liver? Analysing methylation profiles at ASCL2 binding sites (instead of looking for enrichment of ASCL2 binding motives in hypomethylated regions as was already done) might give more insight on genome-wide regulatory effects of this TF especially when comparing HB with fetal cells.

- "Upregulation of the transcription factor ASCL2 raises the possibility that HB might have originated from the immature progenitor cell with a highly proliferative potential similar to the intestinal epithelial cells." – It does, but expression could also be upregulated later on during tumorigenesis (instead of being maintained from progenitor to HB) and I don't think it is feasible to determine this on RNA level. Comparing methylation levels at ASCL2 binding sites (as described above) might give some additional hints.

Some technical questions/comments:

- the average coverage of sequencing libraries (WGS/WGBS) generated by 100bp single reads should be mentioned

- analysis of Illumina Infinium analysis: was the hierarchical clustering performed on all CpG sites of the array? Non LUMP estimates, no tissue correction using reference based or reference free deconvolution. There should be additional prior filtering of low quality sites (bead coverage etc.), please outline this more clearly in the methods section

- line 567: please specify which FDR correction methods were used

minor comments:

- order of figure panels, e.g. Figure 2a described in text after Figure 2b and 2d

- in line 491 reference missing

Point-by-point response to Reviewer 1

"In this work, Nagae et al. conduct an integrated characterization of pediatric liver cancers (mostly hepatoblastoma) with a combination of genomic (whole exome/genome sequencing), transcriptomic (RNA-seq) and epigenomic (methylation arrays and WGBS) methods. Using these valuable data sets, they provide a comprehensive description of driver events, mutational processes and molecular subgroups in hepatoblastoma, with a strong focus on DNA methylation changes.

This is overall a well conducted study that will provide important data sets and results to the community. The global genomic characterization is convincing, as well as the description of DNA methylation-based subgroups. However, the focus on ASXL2 transcription factor as a major regulator of hepatoblastoma suggesting a "cell of origin" for these cancers is enigmatic and not justified by the presented data in my opinion (see question 12 below). I also have many questions and concerns regarding the Methods,

that are often described too briefly to allow reproducibility of the results.”

We are very grateful for these thoughtful comments. We have addressed the reviewer’s concerns and have provided explanations for the same below.

1) “The authors should provide a detailed clinical annotation with details for each sample, not just a summary at the cohort scale as in current Sup Table 1. In addition, given that each omic analysis is performed in only a subset of the samples, the authors should indicate what analysis was performed for each sample.”

We thank the reviewer for this suggestion. We agree that the detailed clinical annotation for each case will be useful for the community. We have now provided the details of clinical and molecular information in Supplementary Table 2.

2) “Pediatric HCC usually develop in a context of constitutional liver disease (tyrosinemia, mitochondrial cytopathies...) and often liver cirrhosis. What were the etiologies of the 9 cases in this cohort, and did the authors identify causal germline variants in HCC?”

We thank the reviewer for this suggestion. We agree that the etiology of pediatric HCC is useful information to interpret the data. Having no underlying chronic liver disease (CLD) was a selection criterion of the JPLT-2 study because such HCC cases with constitutional liver diseases were ineligible for this clinical trial. We investigated the clinical information of the 9 pediatric HCC cases again and confirmed no history of CLD. We further analyzed the germline variants of these cases in whole-exome sequencing data focusing on the same list of 40 cancer predisposition genes as we did for the HB cases (described in lines 137-150). Although 5 HB cases had germline pathogenic variants of cancer-predisposing genes, there were no cases with germline pathogenic variants in the pediatric HCC group. This information has now been clarified in Figure 1a.

3) “The Methods section related to variant calling is very brief. In particular the tools for alignment and variant calling from whole exome sequencing data are not mentioned. No reference is provided regarding the GenomonSV tool used for SV calling, so I suppose it has not been published yet? If so, the authors should provide minimal description of how it works and benchmark against peer-reviewed alternative tools.”

We thank the reviewer for this suggestion. We have now provided the additional

explanation in lines 489-495 as follows:

This genotyper used in this study has been used in several studies (Ref #53, #54, #55) including the ICGC Liver cancer paper (Ref #18) and Clinical trials in the University of Tokyo General Hospital (Ref #56). Furthermore, the source code is publicly available (<http://github.com/genome-rcast/karkinos>).

The comparison result between different genotypers has been reported in the supplementary information (Ref #18) ng.3126-S1.pdf, and a brief methodology has been described in the methods sections of the following previous papers.

Ref #18

Totoki Y, Tatsuno K, Covington KR, et al. Trans - ancestry mutational landscape of hepatocellular carcinoma genomes. *Nat Genet.* 2014;46:1267 - 1273.

Ref #53

Nomura M, et al, Distinct molecular profile of diffuse cerebellar gliomas. *Acta Neuropathol.* 2017 Dec;134(6):941-956.

Ref #54

Kakiuchi M, Nishizawa T, Ueda H, et al. Recurrent gain-of-function mutations of RHOA in diffuse-type gastric carcinoma. *Nat Genet.* 2014;46:583-587.

Ref #55

Wang L, et al, Whole-exome sequencing of human pancreatic cancers and characterization of genomic instability caused by MLH1 haploinsufficiency and complete deficiency. *Genome Res.* 2012 Feb;22(2):208-19.

Ref #56

Kohsaka S, et al, Comprehensive assay for the molecular profiling of cancer by target enrichment from formalin-fixed paraffin-embedded specimens. *Cancer Sci.* 2019 Apr;110(4):1464-1479

4) "The authors state that they identified "26 significantly mutated genes". However, they do not explain how recurrence significance was assessed. Did they account for gene size and genomic covariates using established methods like MutSigCV? Or did they look for an enrichment of functionally important mutations? This is an important point as most candidate driver genes in Fig. 1a are mutated in just 3 samples (2.2%) and they contain large genes like TTN, known to be recurrently mutated in any data set because of its size."

We thank the reviewer for this suggestion. The scarcity of somatic mutations in hepatoblastoma is consistent with the prior results. We have recognized that it is difficult to accurately evaluate the statistical significance for the low frequency of

somatic mutation, so-called “the long tails.” Therefore, in Figure 1a, we listed the recurrently mutated genes in the order of the number of the patients with the mutations. We corrected the sentence in the manuscript (line 122).

5) “The authors used a list of 40 cancer predisposition genes to identify putative causal germline variants. However, they do not provide this list nor explain how it was compiled. In addition, since all mutated genes are tumor suppressors, it would be informative to look for somatic second hits (mutation or loss of the wild-type alleles) in matched tumors. In particular, do the 3 patients with focal 5q22.2 deletions harbor germline APC mutations?”

We thank the reviewer for this suggestion on germline variants.

The following 40 genes were analyzed for germline mutations:

APC, ATM, AXIN1, AXIN2, BAP1, BARD1, BMPR1A, BRCA1, BRCA2, BRIP1, CDH1, CDKN2A, CHEK2, CTNNB1, EGFR, EPCAM, FANCM, FH, HNF1A, MLH1, MSH2, MSH6, MUTYH, NBN, NF1, PALB2, PMS2, POLD1, POLE, PTEN, RAD50, RAD51C, RAD51D, RNF43, SMAD4, STK11, TP53, TSC1, TSC2, VHL.

Most of them were selected due to them being well-established hereditary cancer genes. We supplemented them with regulators of the Wnt/ β -catenin signaling (*CTNNB1, AXIN1, AXIN2, RNF43*) and somatic driver genes of adult liver cancer (*HNF1A, TSC1, TSC2*), which were also reported as potential cancer-predisposing genes. We have now included this information in **Supplementary Table 3**.

Among the 5 patients with germline *APC* mutations, two patients (No. 22 and 58) had somatic truncating mutations of *APC*. Germline variants of other genes did not have somatic mutations. Among the 3 patients with focal 5q22.2 deletions, one patient (No. 63) had a germline mutation of *APC*. The deletion affected the wildtype *APC* allele of the patient, thus constituting a somatic second hit to the gene. These sentences have now been included in **lines 143-144 and lines 189-191**.

6) “The authors used deconstructSigs to decompose substitution patterns into known COSMIC mutational signatures. However, they do not indicate what pool of COSMIC signatures was used. Did they consider the complete panel of 30 signatures in version 2 or 49 signatures in version 3? How did they end up with 3 signatures? Were other signatures not present at all in tumors or did they apply a minimal threshold to consider a signature as operative?”

We thank the reviewer for this suggestion on mutation signatures. We used 30 signatures of the COSMIC database version 2 (**added on line 501**). The scarcity of

mutations in hepatoblastoma made signature analysis challenging, even when we analyzed whole genomes of hepatoblastoma. We initially applied deconstructSigs to the substitution patterns of individual tumors. However, output decomposition was dubious when we checked the clinical records of patients and their parents. Furthermore, the output decomposition was diverse and had little coherence between tumors. This issue was apparently caused by the overfitting of a small stochastic fluctuation in substitution patterns to the COSMIC signatures.

To overcome this issue, we took a conservative approach and pooled the substitution patterns of 33 whole-genome data. This approach increased the number of mutations and would make signature decomposition resistant to stochastic fluctuation. Pooling mutations of different tumors was justified by two observations. (1) SignatureAnalyzer, the software for de novo signature extraction, extracted a single mutation signature from 33 tumors. (2) the substitution patterns of 33 tumors had high similarities with each other (median cosine similarity, 0.74). The high similarity of substitution patterns suggests that the same mutation signatures contributed to a similar extent in all tumors. Pooling would enable the robust detection of such signatures. We admit that some tumors may have a small fraction of sample-specific signatures. However, the scarcity of mutations hindered their detection. More WGS of hepatoblastoma and improvements in analytical software will be required to reliably detect such mutation signatures.

The pooled substitution patterns were decomposed into the COSMIC signatures (version 2) in the following way. Users can specify the number of signatures used by deconstructSigs for decomposition. We ran deconstructSigs specifying the number of signatures between 1 and 5, and monitored the root mean squared error (RMSE) of reconstitution. As the number of signatures increased from 1 through 3, RMSE steadily decreased from 5.6%, 3.5%, to 2.4%. However, changes in RMSE brought by 4th or 5th signatures were less than 1%. We considered that the accuracy of decomposition reached a plateau when three signatures were used. This information has now been included in lines 502-504 of the revised manuscript.

7) “Four subtypes are defined based on transcriptomic profiles. However, no details are provided regarding how the clustering was obtained (gene selection, distance metric, clustering method...), precluding reproducibility of the classification. Also, the authors should assess the stability of the clusters (e.g. by consensus clustering) and explain why they chose the number of 4 clusters. Importantly, several classifications of HB were previously described, in particular the C1/C2 related to prognosis (Cairo et al., Cancer

Cell 2008). The authors should compare their classification with previous work to determine if their groups match previously defined subtypes or define new molecular entities. Finally, it is unclear to me why the third group is called "Mesenchymal", although it mostly shows an overexpression of immune genes?"

We really thank the reviewer for this suggestion on the robustness of subclassification and the consistency with the previous study. We performed the consensus clustering analysis of the gene expression profiles in our cohort and added the results in **Supplementary Figure 6**. As compared with the past result of hierarchical clustering, the "mesenchymal" subtype must be very robust. However, the other three subtypes have sometimes fluctuated between different parameters and methods. We evaluated the consensus matrix by 1000 times of permutation and concluded that it was most robust when $k = 4$ (including the subtype of normal tissue) by the consensus CDF, delta area, and tracking plot.

To compare with the previous work (Cairo et al., Cancer Cell 2008), we used the defined "16 gene set" to re-analyze our data set. As shown in **Supplementary Figure 7c**, the "proliferative" subtype is well consistent with the C2 that was defined in Cairo's paper.

Regarding the "mesenchymal" subtype, we named it by the gene ontology analysis of the upregulated gene sets. As shown in **Supplementary Table 4**, the most significantly enriched GO terms of the upregulated genes in this subtype are "cell adhesion," "extracellular matrix organization," "inflammatory response," and "angiogenesis", those are all typical features of the mesenchymal subtype of epithelial malignancy.

We corrected these results of gene expression analysis in **Supplementary Figure 6** and in **lines 218-238 of the main text** and added the method in **lines 588-596**.

8) "In Sup. Fig. 7, does the heatmap (bottom panel) correspond to methylation levels assessed by microarray? It is not clear from the legend. Same question for Fig. 3. Also no legend is provided in Fig. 3 regarding age, UPD/LOH and IGF2 annotations. The analysis of IGF2 promoters is interesting but the conclusion is a bit enigmatic to me. In Fig. 3, we can see that Pr1 (adult promoter) becomes unmethylated and Pr2 (fetal promoter) becomes methylated with age in normal liver, as expected. Some HB (mostly older) display Pr1 demethylation, whereas the methylation level of Pr2 is heterogeneous. Sup. Fig. 7 shows detailed methylation patterns of the promoters with WGBS. A fetal liver sample (of what age?) is shown as reference but no adult liver. Overall, it is difficult to conclude from these data that HB keep a fetal promoter usage for IGF2."

We thank the reviewer for the constructive question about *IGF2* methylation. In **Sup**

Fig. 8 of the revised manuscript (Sup. Fig. 7 of the 1st submission), the value of Infinium methylation microarray is visualized as a heatmap. We added the bar to clarify the platform to analyze the methylation analysis, similar to **Supplementary Figure 9**. These annotations are also explained in the legend of Figure 3.

We agree that the epigenetic switching of *IGF2* promoters between fetal and adult liver was unclear because we did not show the reference methylation status of adult human liver. As shown in **Supplementary Figure 10**, we added the methylation statuses of this locus in human adult liver data available at the IHEC Epigenome data portal. The revised figure clearly shows the changing of *IGF2* promoter methylation status during liver development. This information has now been included in **lines 250-258**.

9) "According to the main text, Sup. Fig. 9 represents a classification based on the 5,000 most variant probes located in CpG islands, whereas the figure legends mentions a classification based on promoter regions. How were these CpG sites selected? Indeed, only a fraction of promoters contain CpG islands, and a large proportion of CpG islands are not located at promoters..."

We thank the reviewer for this question on the probe selection of promoter-based methylation clustering in **Supplementary Fig 12** (Sup. Fig. 9 in the 1st submission). For this promoter-oriented methylation analysis, we used the method followed for the CpG island methylator phenotype in TCGA studies. First, we removed the probes either designed on CH sites or located on Chr X and Y, and then extracted the probe near the transcription start sites within a distance of less than 1500 bp. To focus on promoter hypermethylation in HB, we removed the probes hypermethylated in normal liver tissues (greater than 0.2 on average). Finally, we used the most variable 5000 probes for the clustering analysis. This probe selection workflow is summarized in **Supplementary Fig 11**. The percentage of probes located at CpG islands (CpGI) and CpGI shores (within 3kb from CpGI) were 63.1% and 29.7%, respectively (**Supplementary Fig 12a**). These loci were overlapped with active or bivalently marked promoters in embryonic stem cells or endoderm cells but became heterochromatin regions in HepG2 cells (**Supplementary Fig 12b**). That might be almost similar to the distribution of human promoters. This information has now been included in **lines 259-269**.

10) "Could the authors comment on the overlap between gene expression and DNA methylation-based classifications? It seems to me that they are not strongly correlated, except maybe the "normal-like" groups? How to the authors interpret this lack of association?"

We thank the reviewer for this question on the relationship between gene expression and DNA methylation-based classification. We performed pairwise comparisons (a comparison of proportions of these 2 categorical variables) by using Fisher's exact test. As shown in Supplementary Figure 13, the subtype “proliferative” of gene expression is significantly overlapped with the subtype E2 of enhancer methylation. This information has now been included in lines 287-289.

11) “The Methods section for motif analysis of differentially methylated enhancers is missing. What regions were considered and what did they use as genomic background? Fig. 4b is hard to understand. What does “Matrix_name” stand for and the “V\$” or “F\$” before transcription factor names? The authors mention a hypermethylation of SP1 motifs in E1 but I do not see SP1 among the top transcription factors in Fig. 4b.”

We thank the reviewer for this question on the motif analysis of differentially methylated enhancers. Regarding the enhancer highly methylated in E1, we described “SP1” by mistake, so we removed it (line 293). We added the detailed information in the method section, lines 598-605, and the Figure legend, lines 742-743 as follows:

We selected the probe sets of “E1 high (n=136),” “E2 low (n=157),” “E2/3 low (n=55),” and “E4 low (n=84)” for the motif analysis to predict the binding of specific transcription factors. We extracted the sequence around the probes (800 bp) and analyzed the enrichment of TF recognition motifs using Transcription factor Affinity Prediction (TRAP) Web Tools (<http://trap.molgen.mpg.de/cgi-bin/home.cgi>). We used the multiple sequence mode with the reference of the “transfac_2010/1_all_matrices” and the background model of human promoters. Multiple test correction was performed using the Benjamini-Hochberg method. The matrix entries have an identifier that indicates one of the six groups of biological species (V\$, vertebrates; I\$, insects; P\$, plants; F\$, fungi; N\$ nematodes; B\$, bacteria), followed by an acronym for the factor the matrix refers to, and a consecutive number discriminating between different matrices for the same factor (http://genexplain.com/wp-content/uploads/2016/09/transfac_documentation_2012-03.pdf).

12) “I do not understand why the authors dedicate a specific focus to ASCL2 transcription factor. It does not appear in the top of the list in Fig. 4b, and I suppose there are many transcription factors upregulated in hepatoblastoma. I do not see why the parallel with intestinal crypts would be relevant for hepatoblastoma. ASCL2 seems to be outside the minimal region of gain in Fig. 5c. The authors do observe methylation

changes in ASCL2 region but I do not see how they can causally link these changes with hepatoblastoma pathogenesis, nor why they link them with "classical HB" in the abstract. Overall, this part seems largely correlative and speculative to me."

We thank the reviewer for asking about *ASCL2*. There are several reasons for focusing on this transcription factor.

- *ASCL2* is an E-box transcription factor that was suggested as a candidate by motif analysis of HB-specific hypomethylation (Fig. 4b).
- Upregulated expression of *ASCL2* is a common feature of clinical HB tissues (Fig. 5).
- *ASCL2* is located at the vicinity of the *IGF2* locus at chromosome 11p, where chromosomal alterations such as UPD or LOH are frequently shown in HB (Fig. 2c).
- *ASCL2* is reported as an essential regulator of the Wnt signal in intestinal cells and the constitutive activation of this pathway due to *CTNNB1* mutation is also a common feature of HB.

Upon the observation of the result of motif analysis (Fig. 4b), the motifs enriched around "E2-low" regions appear to be listed into the two groups, metabolic or hepatic regulators (*DR1*, *PPAR*, *COUP*, *HNF4*) and E-box containing TFs (*EBF1*, *E2A/TCF3*, *MYOGENIN*, *NEUROD1*, *E47/TFE2*), *LMO2COM/BACH2*). The former group is very reasonable because its members are well known as master regulators of mature hepatocytes. However, we do not know the reason why the motifs of the latter E-Box containing TFs are significantly enriched. Most are mainly expressed in neuronal or muscular cells, and not in endodermal lineage cells. However, *ASCL2* was highly expressed in HBs, even if not expressed in the fetal and adult liver.

As the reviewer mentioned, *ASCL2* is located outside the minimal gain region in Fig 5c. But the *ASCL2* enhancing lncRNA, *WINTRLINC1* is located within this region. We agree with the reviewer's comment that the functional roles of *ASCL2* in hepatoblastoma tumorigenesis remain fully clarified at present. However, we confirmed the specific binding of *ASCL2* in this locus (Fig. 5e) and the Wnt-targeted genes (Supplementary Figure. 18c) with local hypomethylation. Comparative analysis of gene expression and promoter methylation suggested a negative correlation between *ASCL2* expression and promoter methylation in *ASCL2* and *WINTRLINC1* (Supplementary Figure. 17). These results might imply the importance of *ASCL2* and *WINTRLINC1* for Wnt-signaling in hepatoblastoma. We added the detailed information in the main text in lines 294-299, 303-343.

13) *“The authors identify several clinical and molecular features associated with poor event-free survival. However, significant molecular markers (CIN and methylation groups) are also related to age. Are these markers significantly associated with EFS independently from age? Regarding methylation of DLX6AS, what beta-value threshold was applied to classify tumors as “M” or “U”, and how was it chosen? DLXAS methylation status (Fig. 6c) seems to largely reproduce the age classification (Fig. 6a). Don’t we get the same curve by simply classifying patient below / above 2 years old? Also, DLXAS seems largely unmethylated in infants (P3, P4) according to Fig. 6d. So how did the authors define an “M” and “U” group among infants. Did they modify the beta value threshold, and how?”*

We thank the reviewer for this question. We corrected the gene name from “DLX6AS” to “DLX6-AS1” following the recent literature. We agree that the age of patients is a major confounding factor of DLX6-AS1 hypermethylation. DLX6-AS1 methylation status seems to largely reproduce the age classification. As shown in **Figure 6a**, infant cases (age < 24 months) showed favorable prognosis than child cases (age: 24-96 months). **Figure 6c** indicates that DLX6-AS1 methylation may be a good predictor to extract patients with poor outcomes in such clinically favorable infant cases.

Regarding the beta-value threshold of DLX6-AS1 methylation, we analyzed the histogram of total data and then set the discriminating cut-off to 0.4 by the probability density estimated via Kernel density estimation (**Supplementary Figure 21**).

14) *“In the Discussion, the authors state that mutational signature 18 is “unique to childhood malignancies”. This is not true as this signature was found in many adult cancers (Alexandrov et al., Nature 2020).”*

We thank the reviewer’s interest in Signature 18. We modified the sentence and added the references as follows: “Signature 18 is a feature that often contributes to childhood malignancies such as neuroblastoma.” in lines **385-386 with the following references:**

Ref #41

Anton G Henssen, Alex Kentsis, Emerging functions of DNA transposases and oncogenic mutators in childhood cancer development, JCI Insight 2018 Oct 18;3(20):e123172

Ref #42

Mia Petljak, Peter J Campbell, Michael R Stratton et al. Characterizing Mutational Signatures in Human Cancer Cell Lines Reveals Episodic APOBEC Mutagenesis, Cell. 2019 Mar 7;176(6):1282-1294.e20.

Point-by-point response to Reviewer 2, expert in methylation and epigenomics

“This is a comprehensive molecular study that integrated genomic, transcriptome and epigenetic profile of a large cohort of hepatoblastoma. Their findings were largely consistent with previously published genomic studies. They showed an overall low mutation rate with CTNNB1 and TERT being the most commonly mutated genes, and frequent dysregulation of chromosome 11p. They also identified molecular subtypes stratified by gene expression profiles and epigenetic profiles and their association with clinical outcome of the patients. The study is well designed and the data is well presented.”

We are very grateful for these thoughtful comments. We confirmed the specific binding of ASCL2 around the *WINTRLINC1* region in a hepatoblastoma cell line, HepG2, which overlapped with the hypomethylated region in HB clinical samples.

“The finding of the upregulation of E-box recognizing transcription factor ASCL2 is potentially interesting. However, detail functional studies on the potential role of ASCL2 in hepatoblastoma and the interaction with Wnt signaling are lacking. Thus, the functional studies of ASCL2 is strongly advised to support the functional importance of ASCL2.”

We thank the reviewer for suggesting the functional studies of ASCL2 in hepatoblastoma. We performed ChIP-seq against endogenous ASCL2 using two hepatoblastoma cell lines, HepG2 and Huh6 to validate the genome-wide binding of this transcription factor. We confirmed that ASCL2 directly binds to the ASCL2 enhancer region (Figure 5e). The genome-wide binding regions are well overlapped with active chromatin regions (H3K4me3- or H3K27ac-marked regions) in HepG2 (Supplementary Figure 18a). To elucidate the functional interaction with canonical Wnt-signaling, we also performed ChIP-seq against CTNNB1 using HepG2 and confirmed the co-localization of ASCL2 and CTNNB1 on this locus. Similar interactions were observed at the other Wnt-targeted genes (Supplementary Figure 18c), strongly suggesting the regulatory contribution to Wnt signaling in HB. This sentence has now been included in lines 330-338.

“The author identified hypermethylation of DLX6AS as single most significant prognostic classifier associated with poorer prognosis. Is this purely as an association or has functional significance? Would the hypermethylation of DLX6AS associated with down-regulation in hepatoblastoma? This should be demonstrated. It has been reported in previous studies that long noncoding RNA DLX6AS promoted tumorigenesis and

cancer progression in several cancer types, e.g. breast cancer and esophageal cancer. The authors shall address the discrepancy to previous findings.”

We are grateful for the critical comment on *DLX6-AS1*. We corrected the description of this gene from *DLX6AS* to *DLX6-AS1*, following recent literature. We selected the hypermethylation of *DLX6-AS1* as the most useful marker to distinguish P1 and P2 from P3 and P4. As the reviewer mentioned, *DLX6-AS1* was reported to be associated with the progression of various types of cancers (e.g. breast, lung, stomach, colon, liver) and is expected to be a potential therapeutic target because of its tumor-specific overexpression. This differentially methylated site (cg22421859, chr7:96622043 on hg19) is located at the intergenic/gene body region of *DLX6-AS1*. The methylation level was positively correlated ($R^2 = 0.633$, data not shown) with the expression level in HB tissues. This information has now been included in lines 357, 361-362, 428-431.

“A number of large scale genomic studies on hepatoblastoma have been published previously. The authors should include a comparison of their finding with published data in the discussion. For example, NFE2L2 is found to be frequently mutated in other studies but seems no mutation was detected in the current study.”

We thank the reviewer for this suggestion. Consistent with the prior result, *CTNNB1* is the most frequently mutated gene in every type of hepatoblastoma and *TERT* promoter mutations are predominantly observed in TLCT/HCN-NOS patients. In the previous study (Eichenmuller M et al, J Hepatology 2014), *NFE2L2* was observed in 10% of HBs. However, we did not find cases with *NFE2L2*-mutations in our cohort. This sentence has now been included in lines 135-136.

Point-by-point response to Reviewer 3, expert in epigenomics:

“Nagae et al. performed a deep genetic, transcriptional and epigenetic analysis of 163 (154 hepatoblastomas and 9 hepatocellular carcinoma) samples. They gather and interpret an impressive number of different genetic, transcriptional and epigenetic. They use these comprehensive data for an integrated analysis combining the diversity in the genetic background (driver, somatic and germline mutations), a classification according to tumor specific expression profiles (4 major groups) and DNA-methylation based grouping based on enhancer signatures. They finally extend their analysis towards a specific set of tumors with LOH/LOI on 11p i.e. subsets of HB with imprinted deregulation. Finally they combine their molecular groupings to stratify the relative survival in relation to genetic and epigenetic data.

The impressive data provide a deep insight in pediatric HB (and HCC) and represent a very rich and unique resource of pediatric liver cancers. In their first chapter describing the “landscape of germline and somatic mutations” the authors find a relatively low number of somatic and germ line mutations most of them occurring in known WNT and TERT related genes including APC (germline). These mutational signatures were neither correlated to age nor birth weight. Using arrays and sequencing they further identified a series of frequent structural variants, copy number alterations and signatures of genome instability – these classifications are later of relevance. A following transcriptional classification into four major subtypes identifies mesenchymal, proliferative, normal and classical signatures each with a relative dominant expression of either cell cycle regulators, WNT signaling genes, immune-specific signatures or normal hepatospecific profiles, while hepatic genes are generally downregulated in all four subtypes.

Finally the authors screen for DNA-methylation changes using 450K arrays and for a subset of 21 cases with either LOH or LOI on chromosomes 11p using WGBS. To identify DNA methylation dependent subclusters they focus on promoters and enhancers (identified in ENCODE datasets). They identify 4 Promoter and 4 Enhancer “classes” and subsequently focus their interest of the enhancer classes to concentrate on binding sites for ASCL2. Their analyse of the WGBS data identifies hypomethylation signatures in certain fetal IGF2 promoter(s) concomitant with a ASCL2 upregulation. They claim that this supports the hypothesis that a distinct class of HB develops from fetal (i.e. intestinal epithelial cell) progenitors.

Finally they use the classified molecular signatures to stratify molecular changes in HB and identify chromosomal instabilities and methylation signatures as major predictors for the outcome. They show predictive signatures in 10 genes with DLX6AS as a most

prominent example.

My major criticism of the paper is that text is difficult to read and that the main messages are not well worked out. Furthermore the authors spike the text with (too) many abbreviations. In the methylome analysis I do not understand the exclusive focus in enhancer Signatures. The authors should show a comprehensive 450K analysis also to rationalize why they focus on enhancers. Furthermore it's unclear why they do not apply tools to estimate the immune cell type contributions as e.g. the LUMP assay (integrated in RnBeads and other packages) or apply deconvolution tools to define covariant CpGs."

We are very grateful for these thoughtful comments.

In the main text, figure/table, and the supplementary information, we used **many abbreviations**; most of them are common in the community of cancer genomics and epigenomics. We added the list of abbreviations to be reader-friendly.

To estimate the immune cell type contributions of clinical tissues used in this study, we performed a LUMP assay using the R package. The results were concordant with the subtyping of methylation (**Figure 4a and Supplementary Figure 12c**).

Major comments:

1) "The main genetic conclusions are extremely difficult to extract for the reader some descriptions are rather detailed and the text is full of abbreviations, brackets etc. disturbing the reading flow and blurring main messages. This part requires a much more clear structure."

We thank the reviewer for this suggestion.

We add the **abbreviation list** in **Supplementary Table 5**.

2) "Gene expression:

- line 219: how were the subtypes annotated (classical, mesenchymal etc.?) or are they simply termed like that? This should be described more clearly.

We thank the reviewer for this suggestion. We added the gene ontology analysis to explain the feature of the expression-based subtypes (**Supplementary Table 4**). This sentence has been included in **lines 218-230**.

- here a differential expression analysis between the subtypes would be interesting to find differential genes with statistical significance"

We thank the reviewer for this suggestion. We selected the differential expression

analysis between the subtypes by the fold change (more than 2.0 or less than 0.5) and statistical significance (p-value < 0.01, by Student's *t*-test). We added the gene list of the differential expression analysis with the 25 largest fold-changes in each subtype in Supplementary Figure 8. This sentence has now been included in lines 230-231.

3) *“Epigenetics:*

- before describing the enhancer specific focus and locus-specific findings (e.g. H19) it would be interesting to obtain information about overall/genome-wide findings from WGBS and the Infinium analysis. How do 450K and WGBS data correlate?

We thank the reviewer for this suggestion. We analyzed the global correlation of each methylation profile between 450K and WGBS. Comparison analysis of both platforms shows a good correlation ($R^2 = 0.951 \pm 0.004$) at the CpG sites with the coverage depth of WGBS being more than 10. This sentence has now been included in lines 247-248, 567-569.

- how were the described loci/genes selected? In the method section I only found a note on hierarchical clustering. A differential methylation analysis between groups or regression analysis in relation to age would make sense to get a genome-wide impression and not just glimpses at individual sites. The volcano plot in Figure 6D indicates that a differential methylation analysis between different ages was already done. However, I cannot find it in the method section and it should be described more clearly in the results part.

We thank the reviewer for this question. For the promoter-oriented methylation analysis, we followed the method used for the CpG island methylator phenotype in TCGA studies. First, we removed the probes either designed on CH sites or located on Chr X and Y, and then extracted the probe near the transcription start sites within a distance of less than 1500 bp. To focus on promoter hypermethylation, we removed the probes hypermethylated in normal liver tissues (greater than 0.2 on average). Finally, we used the most variable 5000 probes for the clustering analysis.

For the enhancer-oriented methylation analysis, we extracted the probes overlapping with the hepatic enhancer regions defined by H3K27ac ChIP-seq data of adult (ENCODE, GSM1112808, GSM1112809) and fetal livers (ENCODE GSM2343044), and then, selected the 1500 most variant probes. This sentence has now been included in lines 260-266 and 279-282 and this probe selection workflow is summarized in Supplementary Figure 11.

- to me the finding that methylation patterns at the IGF2 promoter resembles immature fetal cells on its own is not conclusive enough as it is a single locus. how similar are methylation patterns of HB and fetal liver cells on a genome-wide scale? The 'memory' of a fetal hepatoblast origin should be detectable in more regions."

We thank the reviewer for this suggestion. Numerous differentially methylated regions between adult and fetal livers are observed genome-wide and are partially related to ASCL2 and CTNNB1 binding. We performed the comparative analysis of fetal and adult livers in a genome-wide manner and extracted 4,683 differential methylations. As shown in the newly added **Supplementary Figure 20**, among the 3351 CpG sites hypermethylated in the adult livers, 53% showed sustained hypomethylation in HB, including *IGF2*, *ESR1*, *NR4A2*. On the contrary, among the 1332 CpG sites hypermethylated in the fetal liver, 19% showed sustained hypermethylation. This sentence has now been included in lines **338-342** and the genes have been added to the gene list of the sustained fetal-liver-like differential methylation in **Supplementary Figure 20**.

4) ASCL2

- line 302: is this upregulation also detected in the HB RNA-seq data?

We thank the reviewer for this suggestion. There are no cases with local amplification in HB samples. HB shows upregulation of ASCL2 without gene amplification but through epigenetic activation and auto-regulation by the positive feedback loop. We emphasized this observation on **line 343**.

- Is it known whether ASCL2 preferentially binds to unmethylated/methylated binding sites? How is the methylation status at ASCL2 binding sites in HB and fetal liver? Analysing methylation profiles at ASCL2 binding sites (instead of looking for enrichment of ASCL2 binding motives in hypomethylated regions as was already done) might give more insight on genome-wide regulatory effects of this TF especially when comparing HB with fetal cells.

We thank the reviewer for this suggestion. We analyzed the methylation level of HB and normal liver tissues at ASCL2 binding sites defined by the ChIP-sequencing result of HepG2. As shown in **Supplementary Figure 12b**, ASCL2 binding sites are generally unmethylated in HB. Regarding the co-binding sites of ASCL2 and CTNNB1, the strongly bound regions (ASCL2 ChIP score: > 50) are stably unmethylated among non-cancerous liver tissues but the moderately bound regions (ASCL2 ChIP score: 30-50) show a gradual gain of methylation along with age (**Supplementary Figure 13**).

More functional experiments are needed to know whether ASCL2-binding is sensitive to the CpG methylation around the recognition motifs. This sentence has now been included in lines 330-342.

- “Upregulation of the transcription factor ASCL2 raises the possibility that HB might have originated from the immature progenitor cell with a highly proliferative potential similar to the intestinal epithelial cells.” – It does, but expression could also be upregulated later on during tumorigenesis (instead of being maintained from progenitor to HB) and I don’t think it is feasible to determine this on RNA level. Comparing methylation levels at ASCL2 binding sites (as described above) might give some additional hints.

We thank the reviewer for these thoughtful suggestions. We analyzed the methylation level of ASCL2 and CTNNB1 co-binding sites in human liver tissues. As shown in **Supplementary Figure 19**, weakly bound regions (ChIP score: 30-50) are gradually methylated with age among non-cancerous liver tissues. However, strongly bound regions remained unmethylated among these samples. Further integrated analysis such as ChIP-sequencing of fetal endodermal tissues as well as RNA-sequencing is necessary to resolve this issue.

We corrected the sentence in the abstract and the main text in line 404-405 as follows: Prolonged upregulation of ASCL2, as well as fetal-liver-like methylation patterns of IGF2 promoters, suggests their “cell of origin” derived from the premature hepatoblast, similar to intestinal epithelial cells, which are highly proliferative.

Some technical questions/comments:

- the average coverage of sequencing libraries (WGS/WGBS) generated by 100bp single reads should be mentioned

We thank the reviewer for this suggestion. We added the average coverage of the sequencing libraries for WGS and WGBS in **lines 487 and 567**, respectively.

- analysis of Illumina Infinium analysis: was the hierarchical clustering performed on all CpG sites of the array? Non LUMP estimates, no tissue correction using reference based of reference free deconvolution. There should be additional prior filtering of low quality sites (bead coverage etc.), please outline this more clearly in the methods section

We thank the reviewer for this very important suggestion. For the promoter-oriented methylation analysis, we removed the probes hypermethylated in normal liver tissues (greater than 0.2 on average). For the enhancer-oriented methylation analysis, we

extracted the probes overlapping with the hepatic enhancer regions defined by the H3K27ac ChIP-seq data of adult (ENCODE, GSM1112808, GSM1112809) and fetal livers (ENCODE GSM2343044). Both filtering processes might be effective in removing the hypermethylated or hypomethylated regions due to the infiltration of non-hepatocyte cells.

In addition, we analyzed the LUMP estimate of clinical HB tissues using InfiniumPurify (<https://github.com/Xiaoqizheng/InfiniumPurify>). As shown in **Supplementary Fig 12c**, the P4 (Normal like) subtype show low values of tumor purity. Similarly, the E4 enhancer methylation subtype in Figure 4a shows a low rate of tumor purity. Therefore, we have avoided mentioning the molecular feature of cancer cells in these cancer tissues. This sentence has been included in **lines 274-275**.

- line 567: please specify which FDR correction methods were used

We thank the reviewer for this suggestion. We performed FDR correction by Benjamini-Hochberg method in **Figure 6d**. This sentence is added in the methods section on **line 654**.

minor comments:

- order of figure panels, e.g. Figure 2a described in text after Figure 2b and 2d

We thank the reviewer for this suggestion. We corrected the order of the figure panels of Figure 2 in the text.

- in line 491 reference missing

We thank the reviewer for this suggestion. We added the reference for the method to detect *CTNNB1* mutation by Sanger sequencing on line **532**.

REVIEWERS' COMMENTS

Reviewer #1 (Remarks to the Author):

The authors have significantly improved their manuscript in this revised version and provided satisfactory answers to all the concerns raised in my initial review.

Reviewer #2 (Remarks to the Author):

This is a comprehensive analysis that integrated genomic, epigenomic and transcriptomic analysis in a large cohort of paediatric hepatoblastomas. In concordance with previous studies, they found low mutation rate in Pediatric hepatoblastomas with CTNNB1 and TERT being the most commonly mutated genes and frequent dysregulation of chromosome 11p. By using genomic, transcriptomic and epigenomic data they classified the hepatoblastomas into distinct subgroups with the aim of identification of molecular signature for risk stratification. The study was well planned and the data generated provide valuable information for the understanding of genetic landscape of Pediatric hepatoblastoma. This is a revised manuscript. The comments from previous review have been addressed adequately.

Reviewer #3 (Remarks to the Author):

no further requests